# Towards Understanding the Data Dependency of Mixup-style Training

**Muthu Chidambaram**[1]**, Xiang Wang**[1]**, Yuzheng Hu**[2]**, Chenwei Wu**[1]**, and Rong Ge**[1]

[1]Duke University, [2]University of Illinois at Urbana-Champaign

## Abstract

In the Mixup training paradigm, a model is trained using convex combinations of data points and their associated labels. Despite seeing very few true data points during training, models trained using Mixup seem to still minimize the original empirical risk and exhibit better generalization and robustness on various tasks when compared to standard training. In this paper, we investigate how these benefits of Mixup training rely on properties of the data in the context of classification. For minimizing the original empirical risk, we compute a closed form for the Mixup-optimal classification, which allows us to construct a simple dataset on which minimizing the Mixup loss can provably lead to learning a classifier that does not minimize the empirical loss on the data. On the other hand, we also give sufficient conditions for Mixup training to also minimize the original empirical risk. For generalization, we characterize the margin of a Mixup classifier, and use this to understand why the decision boundary of a Mixup classifier can adapt better to the full structure of the training data when compared to standard training. In contrast, we also show that, for a large class of linear models and linearly separable datasets, Mixup training leads to learning the same classifier as standard training.

## 1 Introduction

Mixup (Zhang et al., 2018) is a modification to the standard supervised learning setup which involves training on convex combinations of pairs of data points and their labels instead of the original data itself. In the original paper, Zhang et al. (2018) demonstrated that training deep neural networks using Mixup leads to better generalization performance, as well as greater robustness to adversarial attacks and label noise on image classification tasks. The empirical advantages of Mixup training have been affirmed by several follow-up works (He et al., 2019; Thulasidasan et al., 2019; Lamb et al., 2019; Arazo et al., 2019; Guo, 2020). The idea of Mixup has also been extended beyond the supervised learning setting, and been applied to semi-supervised learning (Berthelot et al., 2019; Sohn et al., 2020), contrastive learning (Verma et al., 2021; Lee et al., 2020), privacy-preserving learning (Huang et al., 2021), and learning with fairness constraints (Chuang & Mroueh, 2021).

However, from a theoretical perspective, Mixup training is still mysterious even in the basic multi-class classfication setting – why should the output of a linear mixture of two training samples be the same linear mixture of their labels, especially when considering highly nonlinear models? Despite several recent theoretical results (Guo et al., 2019; Carratino et al., 2020; Zhang et al., 2020; 2021), there is still not a complete understanding of why Mixup training actually works in practice. In this paper, we try to understand *why* Mixup works by first understanding *when* Mixup works: in particular, how the properties of Mixup training rely on the structure of the training data.

We consider two properties for classifiers trained with Mixup. First, even though Mixup training does not observe many original data points during training, it usually can still correctly classify all of the original data points (empirical risk minimization (ERM)). Second, the aforementioned empirical works have shown how classifiers trained with Mixup often have better adversarial robustness and generalization than standard training. In this work, we show that both of these properties can rely heavily on the data used for training, and that they need not hold in general.

**Main Contributions and Related Work.** The idea that Mixup can potentially fail to minimize the original risk is not new; Guo et al. (2019) provide examples of how Mixup labels can conflict with actual data point labels. However, their theoretical results do not characterize the data and

---

Correspondence to Muthu Chidambaram (`muthu@cs.duke.edu`).

model conditions under which this failure can provably happen when minimizing the Mixup loss. In Section 2 of this work, we provide a concrete classification dataset on which continuous approximate-minimizers of the Mixup loss can fail to minimize the empirical risk. We also provide sufficient conditions for Mixup to minimize the original risk, and show that these conditions hold approximately on standard image classification benchmarks.

With regards to generalization and robustness, the parallel works of Carratino et al. (2020) and Zhang et al. (2020) showed that Mixup training can be viewed as minimizing the empirical loss along with a data-dependent regularization term. Zhang et al. (2020) further relate this term to the adversarial robustness and Rademacher complexity of certain function classes learned with Mixup. In Section 3, we take an alternative approach to understanding generalization and robustness by analyzing the margin of Mixup classifiers. Our perspective can be viewed as complementary to that of the aforementioned works, as we directly consider the properties exhibited by a Mixup-optimal classifier instead of considering what properties are encouraged by the regularization effects of the Mixup loss. In addition to our margin analysis, we also show that for the common setting of linear models trained on high-dimensional Gaussian features both Mixup (for a large class of mixing distributions) and ERM with gradient descent learn the same classifier with high probability.

Finally, we note the related works that are beyond the scope of our paper; namely the many Mixup-like training procedures such as Manifold Mixup (Verma et al., 2019), Cut Mix (Yun et al., 2019), Puzzle Mix (Kim et al., 2020), and Co-Mixup (Kim et al., 2021).

## 2 MIXUP AND EMPIRICAL RISK MINIMIZATION

The goal of this section is to understand when Mixup training can also minimize the empirical risk. Our main technique for doing so is to derive a closed-form for the Mixup-optimal classifier over a sufficiently powerful function class, which we do in Section 2.2 after introducing the basic setup in Section 2.1. We use this closed form to motivate a concrete example on which Mixup training does not minimize the empirical risk in Section 2.3, and show under mild nondegeneracy conditions that Mixup will minimize the empirical risk in Section 2.4.

### 2.1 SETUP

We consider the problem of $k$-class classification where the classes $1, ..., k$ correspond to compact disjoint sets $X_1, ..., X_k \subset \mathbb{R}^n$ with an associated probability measure $\mathbb{P}_X$ supported on $X = \bigcup_{i=1}^k X_i$. We use $\mathcal{C}$ to denote the set of all functions $g : \mathbb{R}^n \to [0, 1]^k$ satisfying the property that $\sum_{i=1}^k g^i(x) = 1$ for all $x$ (where $g^i$ represents the $i$-th coordinate function of $g$). We refer to a function $g \in \mathcal{C}$ as a *classifier*, and say that $g$ classifies $x$ as class $j$ if $j = \text{argmax}_i g^i(x)$. The cross-entropy loss associated with such a classifier $g$ is then:

$$J(g, \mathbb{P}_X) = -\sum_{i=1}^k \int_{X_i} \log g^i(x) d\mathbb{P}_X(x)$$

The goal of standard training is to learn a classifier $h \in \text{argmin}_{g \in \mathcal{C}} J(g, \mathbb{P}_X)$. Any such classifier $h$ will necessarily satisfy $h^i(x) = 1$ on $X_i$ since the $X_i$ are disjoint.

**Mixup.** In the Mixup version of our setup, we are interested in minimizing the cross-entropy of convex combinations of the original data and their classes. These convex combinations are determined according to a probability measure $\mathbb{P}_f$ whose support is $[0, 1]$, and we assume this measure has a density $f$. For two points $s, t \in X$, we let $z_{st}(\lambda) = \lambda s + (1 - \lambda)t$ (and use $z_{st}$ when $\lambda$ is understood) and define the Mixup cross-entropy on $s, t$ with respect to a classifier $g$ as:

$$\ell_{mix}(g, s, t, \lambda) = \begin{cases} -\log g^i(z_{st}) & s, t \in X_i \\ -\left(\lambda \log g^i(z_{st}) + (1 - \lambda) \log g^j(z_{st})\right) & s \in X_i, t \in X_j \end{cases}$$

Having defined $\ell_{mix}$ as above, we may write the component of the full Mixup cross-entropy loss corresponding to mixing points from classes $i$ and $j$ as:

$$J_{mix}^{i,j}(g, \mathbb{P}_X, \mathbb{P}_f) = \int_{X_i \times X_j \times [0,1]} \ell_{mix}(g, s, t, \lambda) \, d(\mathbb{P}_X \times \mathbb{P}_X \times \mathbb{P}_f)(s, t, \lambda)$$

The final Mixup cross-entropy loss is then the sum of $J_{mix}^{i,j}$ over all $i, j \in \{1, ..., k\}$ (corresponding to all possible mixings between classes, including themselves):

$$J_{mix}(g, \mathbb{P}_X, \mathbb{P}_f) = \sum_{i=1}^{k} \sum_{j=1}^{k} J_{mix}^{i,j}(g, \mathbb{P}_X, \mathbb{P}_f)$$

**Relation to Prior Work.** We have opted for a more general definition of the Mixup loss (at least when constrained to multi-class classification) than prior works. This is not generality for generality's sake, but rather because many of our results apply to any mixing distribution supported on $[0, 1]$. One obtains the original Mixup formulation of Zhang et al. (2018) for multi-class classification on a finite dataset by taking the $X_i$ to be finite sets, and choosing $\mathbb{P}_X$ to be the normalized counting measure (corresponding to a discrete uniform distribution). Additionally, $\mathbb{P}_f$ is chosen to have density $\text{Beta}(\alpha, \alpha)$, where $\alpha$ is a hyperparameter.

## 2.2 MIXUP-OPTIMAL CLASSIFIER

Given our setup, we now wish to characterize the behavior of a Mixup-optimal classifier at a point $x \in \mathbb{R}^n$. However, if the optimization of $J_{mix}$ is considered over the class of functions $\mathcal{C}$, this is intractable (to the best of our knowledge) due to the lack of regularity conditions imposed on functions in $\mathcal{C}$. We thus wish to constrain the optimization of $J_{mix}$ to a class of functions that is sufficiently powerful (so as to include almost all practical settings) while still allowing for local analysis. To do so, we will need the following definitions, which will also be referenced throughout the results in this section and the next:

$$A_{x,\epsilon}^{i,j} = \{(s,t,\lambda) \in X_i \times X_j \times [0,1] : \ \lambda s + (1-\lambda)t \in B_\epsilon(x)\}$$

$$A_{x,\epsilon,\delta}^{i,j} = \{(s,t,\lambda) \in X_i \times X_j \times [0, 1-\delta] : \ \lambda s + (1-\lambda)t \in B_\epsilon(x)\}$$

$$X_{mix} = \left\{ x \in \mathbb{R}^n : \bigcup_{i,j} A_{x,\epsilon}^{i,j} \text{ has positive measure for every } \epsilon > 0 \right\}$$

$$\xi_{x,\epsilon}^{i,j} = \int_{A_{x,\epsilon}^{i,j}} d(\mathbb{P}_X \times \mathbb{P}_X \times \mathbb{P}_f)(s,t,\lambda)$$

$$\xi_{x,\epsilon,\lambda}^{i,j} = \int_{A_{x,\epsilon}^{i,j}} \lambda \, d(\mathbb{P}_X \times \mathbb{P}_X \times \mathbb{P}_f)(s,t,\lambda)$$

The set $A_{x,\epsilon}^{i,j}$ represents all points in $X_i \times X_j$ that have lines between them intersecting an $\epsilon$-neighborhood of $x$, while the set $A_{x,\epsilon,\delta}^{i,j}$ represents the restriction of $A_{x,\epsilon}^{i,j}$ to only those points whose connecting line segments intersect an $\epsilon$-neighborhood of $x$ with $\lambda$ values bounded by $1 - \delta$ (used in Section 3). The set $X_{mix}$ corresponds to all points for which every neighborhood factors into $J_{mix}$. The $\xi_{x,\epsilon}^{i,j}$ term represents the measure of the set $A_{x,\epsilon}^{i,j}$ while $\xi_{x,\epsilon,\lambda}^{i,j}$ represents the expectation of $\lambda$ over the same set. To provide better intuition for these definitions, we provide visualizations in Section B of the appendix. We can now define the subset of $\mathcal{C}$ to which we will constrain our optimization of $J_{mix}$.

**Definition 2.1.** Let $\mathcal{C}^*$ to be the subset of $\mathcal{C}$ for which every $h \in \mathcal{C}^*$ satisfies $h(x) = \lim_{\epsilon \to 0} \operatorname{argmin}_{\theta \in [0,1]^k} J_{mix}(\theta)|_{B_\epsilon(x)}$ for all $x \in X_{mix}$ when the limit exists. Here $J_{mix}(\theta)|_{B_\epsilon(x)}$ represents the Mixup loss for a constant function with value $\theta$ with the restriction of each term in $J_{mix}$ to the set $A_{x,\epsilon}^{i,j}$.

We immediately justify this definition with the following proposition.

**Proposition 2.2.** Any function $h \in \operatorname{argmin}_{g \in \mathcal{C}^*} J_{mix}(g, \mathbb{P}_X, \mathbb{P}_f)$ satisfies $J_{mix}(h) \le J_{mix}(g)$ for any continuous $g \in \mathcal{C}$.

**Proof Sketch.** We can argue directly from definitions by considering points in $X_{mix}$ for which $h$ and $g$ differ.

Proposition 2.2 demonstrates that optimizing over $\mathcal{C}^*$ is at least as good as optimizing over the subset of $\mathcal{C}$ consisting of continuous functions, so we cover most cases of practical interest (i.e. optimizing deep neural networks). As such, the term "Mixup-optimal" is intended to mean optimal with respect to $\mathcal{C}^*$ throughout the rest of the paper. We may now characterize the classification of a Mixup-optimal classifier on $X_{mix}$.

**Lemma 2.3.** For any point $x \in X_{mix}$ and $\epsilon > 0$, there exists a continuous function $h_\epsilon$ satisfying:

$$h_\epsilon^i(x) = \frac{\xi_{x,\epsilon}^{i,i} + \sum_{j \neq i} \left( \xi_{x,\epsilon,\lambda}^{i,j} + (\xi_{x,\epsilon}^{j,i} - \xi_{x,\epsilon,\lambda}^{j,i}) \right)}{\sum_{q=1}^{k} \left( \xi_{x,\epsilon}^{q,q} + \sum_{j \neq q} \left( \xi_{x,\epsilon,\lambda}^{q,j} + (\xi_{x,\epsilon}^{j,q} - \xi_{x,\epsilon,\lambda}^{j,q}) \right) \right)} \tag{1}$$

With the property that $\lim_{\epsilon \to 0} h_\epsilon(x) = h(x)$ for every $h \in \operatorname{argmin}_{g \in \mathcal{C}^*} J_{mix}(g, \mathbb{P}_X, \mathbb{P}_f)$ when the limit exists.

**Proof Sketch.** We set $h_\epsilon = \operatorname{argmin}_{\theta \in [0,1]^k} J_{mix}(\theta)|_{B_\epsilon(x)}$ and show that this is well-defined, continuous, and has the above form using the strict convexity of the minimization problem.

**Remark 2.4.** For the important case of finite datasets, it will be shown that the limit above always exists as part of the proof of Theorem 3.2.

The expression for $h_\epsilon^i$ just represents the expected location of the point $x$ on all lines between class $i$ and other classes, normalized by the sum of the expected locations for all classes. It can be simplified significantly if $\mathbb{P}_f$ is assumed to be symmetric; we give this as a corollary after the proof in Section C of the Appendix. Importantly, we note that while $h_\epsilon$ as defined in Lemma 2.3 is continuous for every $\epsilon > 0$, its pointwise limit $h$ need not be, which we demonstrate below.

**Proposition 2.5.** Let $X_1 = \{(0,1), (0,-1)\}$ and let $X_2 = \{(1,0), (-1,0)\}$, with $\mathbb{P}_X$ being discrete uniform over $X_1 \cup X_2$ and $\mathbb{P}_f$ being continuous uniform over $[0,1]$. Then the Mixup-optimal classifier $h$ is discontinuous at $(0,0)$.

**Proof Sketch.** One may explicitly compute for $x = (0,0)$ that $h^1(x) = h^2(x) = \frac{1}{2}$.

Proposition 2.5 illustrates our first significant difference between Mixup training and standard training: there *always* exists a minimizer of the empirical cross-entropy $J$ that can be extended to a continuous function (since a minimizer is constant on the class supports and not constrained elsewhere), whereas depending on the data the minimizer of $J_{mix}$ can be discontinuous.

### 2.3 A MIXUP FAILURE CASE

With that in mind, several model classes popular in practical applications consist of continuous functions. For example, neural networks with ReLU activations are continuous, and several works have noted that they are Lipschitz continuous with shallow networks having approximately small Lipschitz constant (Scaman & Virmaux, 2019; Fazlyab et al., 2019; Latorre et al., 2020). Given the regularity of such models, we are motivated to consider the continuous approximations $h_\epsilon$ in Lemma 2.3 and see if it is possible to construct a dataset on which $h_\epsilon$ (for a fixed $\epsilon$) can fail to classify the original points correctly. We thus consider the following dataset:

**Definition 2.6.** [3-Point Alternating Line] We define $\mathcal{X}_3^2$ to be the binary classification dataset consisting of the points $\{0, 1, 2\}$ classified as $\{1, 2, 1\}$. In our setup, this corresponds to $X_1 = \{0, 2\}$ and $X_2 = \{1\}$ with $\mathbb{P}_X = \frac{1}{3} 1_{\{0,1,2\}}$.

Intuitively, the reason why Mixup can fail on $\mathcal{X}_3^2$ is that, for choices of $\mathbb{P}_f$ that concentrate about $\frac{1}{2}$, we will have by Lemma 2.3 that the Mixup-optimal classification in a neighborhood of point 1 should skew towards class 1 instead of class 2 due to the sandwiching of point 1 between points 0 and 2. The canonical choice of $\mathbb{P}_f$ corresponding to a mixing density of $\text{Beta}(\alpha, \alpha)$ is one such choice:

**Theorem 2.7.** Let $\mathbb{P}_f$ have associated density $\text{Beta}(\alpha, \alpha)$. Then for any classifier $h_\epsilon$ on $\mathcal{X}_3^2$ (as defined in Lemma 2.3), we may choose $\alpha$ such that $h_\epsilon$ does not achieve 0 classification error on $\mathcal{X}_3^2$.

**Proof Sketch.** For any $\epsilon > 0$, we can bound the $\xi$ terms in Equation 1 using the fact that $\text{Beta}(\alpha, \alpha)$ is strictly subgaussian (Marchal & Arbel, 2017), and then choose $\alpha$ appropriately.

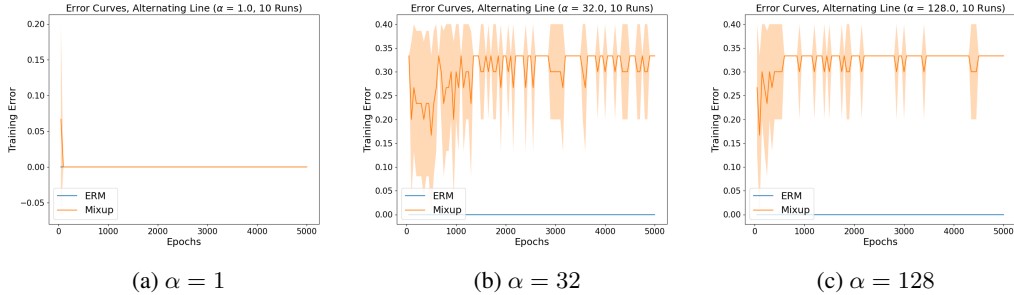

(a) $\alpha = 1$      (b) $\alpha = 32$      (c) $\alpha = 128$

Figure 1: Training error for Mixup and regular training on $\mathcal{X}_3^2$. Each curve corresponds to the mean of 10 training runs, and the area around each curve represents a region of one standard deviation.

**Experiments.** The result of Theorem 2.7 leads us to believe that the Mixup training of a continuous model should fail on $\mathcal{X}_3^2$ for appropriately chosen $\alpha$. To verify that the theory predicts the experiments, we train a two-layer feedforward neural network with 512 hidden units and ReLU activations on $\mathcal{X}_3^2$ with and without Mixup. The implementation of Mixup training does not differ from the theoretical setup; we uniformly sample pairs of data points and train on their mixtures. Our implementation uses PyTorch (Paszke et al., 2019) and is based heavily on the open source implementation of Manifold Mixup (Verma et al., 2019) by Shivam Saboo. Results for training using (full-batch) Adam (Kingma & Ba, 2015) with the suggested (and common) hyperparameters of $\beta_1 = 0.9, \beta_2 = 0.999$ and a learning rate of $0.001$ are shown in Figure 1. The class 1 probabilities for each point in the dataset outputted by the learned Mixup classifiers from Figure 1 are shown in Table 1 below:

| $h$ | 0 | 1 | 2 |
|---|---|---|---|
| $\alpha = 1$ | 0.995 | 0.156 | 0.979 |
| $\alpha = 32$ | 1.000 | 0.603 | 0.997 |
| $\alpha = 128$ | 1.000 | 0.650 | 0.997 |

Table 1: Mixup model evaluations on $\mathcal{X}_3^2$ for different choices of $\alpha$.

We see from Figure 1 and Table 1 that Mixup training fails to correctly classify the points in $\mathcal{X}_3^2$ for $\alpha = 32$, and this misclassification becomes more exacerbated as we increase $\alpha$. The choice of $\alpha$ for which misclassifications begin to happen is largely superficial; we show in Section D of the Appendix that it is straightforward to construct datasets in the style of $\mathcal{X}_3^2$ for which Mixup training will fail even for the very mild choice of $\alpha = 1$. We focus on the case of $\mathcal{X}_3^2$ here to simplify the theory. The key takeaway is that, for datasets that exhibit (approximately) collinear structure amongst points, it is possible for inappropriately chosen mixing distributions to cause Mixup training to fail to minimize the original empirical risk.

## 2.4 Sufficient Conditions for Minimizing the Original Risk

The natural follow-up question to the results of the previous subsection is: under what conditions on the data can this failure case be avoided? In other words, when can the Mixup-optimal classifier classify the original data points correctly while being continuous at those points?

Prior to answering that question, we first point out that if discontinuous functions are allowed, then Mixup training always minimizes the original risk on finite datasets:

**Proposition 2.8.** Consider $k$-class classification where the supports $X_1, ..., X_k$ are finite and $\mathbb{P}_X$ corresponds to the discrete uniform distribution. Then for every $h \in \mathrm{argmin}_{g \in \mathcal{C}^*} J_{mix}(g, \mathbb{P}_X, \mathbb{P}_f)$, we have that $h^i(x) = 1$ on $X_i$.

**Proof Sketch.** Only the $\xi_{x,\epsilon}^{i,i}$ term doesn't vanish in $h_\epsilon^i(x)$ as $\epsilon \to 0$, as the mixing distribution is continuous and cannot assign positive measure to $x$ alone when mixing two points that are not $x$.

Note that Proposition 2.8 holds for **any** continuous mixing distribution $\mathbb{P}_f$ supported on $[0, 1]$ - we just need a rich enough model class.

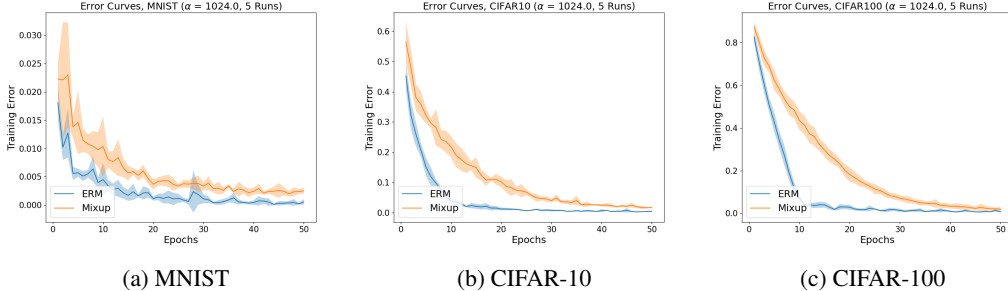

(a) MNIST          (b) CIFAR-10          (c) CIFAR-100

Figure 2: Mean and single standard deviation of 5 training runs for Mixup ($\alpha = 1024$) and ERM on the *original training data*. Mixup achieves near-identical (within 1%) training accuracy to ERM.

In order to obtain the result of Proposition 2.8 with the added restriction of continuity of $h$ on each of the $X_i$, we need to further assume that the collinearity of different class points that occurred in the previous section does not happen.

**Assumption 2.9.** For any point $x \in X_i$, there do not exist $u \in X$ and $v \in X_j$ for $j \neq i$ such that there is a $\lambda > 0$ for which $x = \lambda u + (1 - \lambda)v$.

A visualization of Assumption 2.9 is provided in Section B of the appendix. With this assumption in hand, we obtain the following result as a corollary of Theorem 3.2 which is proved in the next section:

**Theorem 2.10.** We consider the same setting as Proposition 2.8 and further suppose that Assumption 2.9 is satisfied. Then for every $h \in \operatorname{argmin}_{g \in \mathcal{C}^*} J_{mix}(g, \mathbb{P}_X, \mathbb{P}_f)$, we have that $h^i(x) = 1$ on $X_i$ and that $h$ is continuous on $X$.

**Application of Sufficient Conditions.** The practical take-away of Theorem 2.10 is that if a dataset does not exhibit approximate collinearity between points of different classes, then Mixup training should achieve near-identical training error on the *original training data* when compared to ERM. We validate this by training ResNet-18 (He et al., 2015) (using the popular implementation of Kuang Liu) on MNIST (LeCun, 1998), CIFAR-10, and CIFAR-100 (Krizhevsky, 2009) with and without Mixup for 50 epochs with a batch size of 128 and otherwise identical settings to the previous subsection. For Mixup, we consider mixing using $\mathrm{Beta}(\alpha, \alpha)$ for $\alpha = 1, 32, 128,$ and $1024$ to cover a wide band of mixing distributions. Our experimental results for the "worst case" of $\alpha = 1024$ (the other choices are strictly closer to ERM in training accuracy) are shown in Figure 2, while the other experiments can be found in Section D of the Appendix.

We now check that the theory can predict the results of our experiments by verifying Assumption 2.9 approximately (exact verification is too expensive). We sample one epoch's worth of Mixup points (to simulate training) from a downsampled version of each train dataset, and then compute the minimum distances between each Mixup point and points from classes other than the two mixed classes. The minimum over these distances corresponds to an estimate of $\epsilon$ in Assumption 2.9. We compute the distances for both training and test data, to see whether good training but poor test performance can be attributed to test data conflicting with mixed training points. Results are shown below in Table 2.

| Comparison Type | MNIST | CIFAR-10 | CIFAR-100 |
|:---:|:---:|:---:|:---:|
| Mixup/Train | 11.433 | 17.716 | 12.936 |
| Mixup/Test | 11.897 | 22.133 | 15.076 |

Table 2: Minimum Euclidean distance results using our approximation procedure with Mixup points generated using $\mathrm{Beta}(1024, 1024)$. We downsample all datasets to 20%, to compare to the experiments of Guo et al. (2019).

To interpret the estimated $\epsilon$ values in Table 2, we note that unless $\epsilon \ll 1/L$ (where $L$ is the Lipschitz constant of the model being considered), a Mixup point cannot conflict with an original point (since the function has enough flexibility to fit both). Due to the estimated large Lipschitz constants of deep networks (Scaman & Virmaux, 2019), our $\epsilon$ values certainly do not fall in this regime, explaining how the near-identical performance to ERM is possible on the original datasets. We remark that our results challenge an implication of Guo et al. (2019), which was that training/test performance on

the above benchmarks degrades with high values of $\alpha$ due to conflicts between original points and Mixup points.

## 2.5 THE RATE OF EMPIRICAL RISK MINIMIZATION USING MIXUP

Another striking aspect of the experiments in Figure 2 is that Mixup training minimizes the original empirical risk at a very similar rate to that of direct empirical risk minimization. A priori, there is no reason to expect that Mixup should be able to do this - a simple calculation shows that Mixup training only sees one true data point per epoch in expectation (each pair of points is sampled with probability $\frac{1}{m^2}$ and there are $m$ true point pairs and $m$ pairs seen per epoch, where $m$ is the dataset size). The experimental results are even more surprising given that we are training using $\alpha = 1024$, which essentially corresponds to training using the midpoints of the original data points. This seems to imply that it is possible to recover the classifications of the original data points from the midpoints alone (not including the midpoint of a point and itself), and similar phenomena has indeed been observed in recent empirical work (Guo, 2021). We make this rigorous with the following result:

**Theorem 2.11.** Suppose $\{x_1, ..., x_m\}$ with $m \geq 6$ are sampled from $X$ according to $\mathbb{P}_X$, and that $\mathbb{P}_X$ has a density. Then with probability 1, we can uniquely determine the points $\{x_1, ..., x_m\}$ given only the $\binom{m}{2}$ midpoints $\{x_{i,j}\}_{1 \leq i < j \leq m}$.

**Proof Sketch.** The idea is to represent the problem as a linear system, and show using rank arguments that the sets of $m$ points that cannot be uniquely determined are a measure zero set.

Theorem 2.11 shows, in an information-theoretic sense, that it is possible to obtain the original data points (and therefore also their labels) from only their midpoints. While this gives more theoretical backing as to why it is possible for Mixup training using $\mathrm{Beta}(1024, 1024)$ to recover the original data point classifications with very low error, it does not explain why this actually happens in practice at the rate that it does. A full theoretical analysis of this phenomenon would necessarily require analyzing the training dynamics of neural networks (or another model of choice) when trained only on midpoints of the original data, which is outside the intended scope of this work. That being said, we hope that such analysis will be a fruitful line of investigation for future work.

## 3 GENERALIZATION PROPERTIES OF MIXUP CLASSIFIERS

Having discussed how Mixup training differs from standard empirical risk minimization with regards to the original training data, we now consider how a learned Mixup classifier can differ from one learned through empirical risk minimization on unseen test data. To do so, we analyze the *per-class margin* of Mixup classifiers, i.e. the distance one can move from a class support $X_i$ while still being classified as class $i$.

### 3.1 THE MARGIN OF MIXUP CLASSIFIERS

Intuitively, if a point $x$ falls only on line segments between $X_i$ and some other classes $X_j, ...,$ and if $x$ always falls closer to $X_i$ than the other classes, we can expect $x$ to be classified according to class $i$ by the Mixup-optimal classifier due to Lemma 2.3. To make this rigorous, we introduce another assumption that generalizes Assumption 2.9 to points outside of the class supports:

**Assumption 3.1.** For a class $i$ and a point $x \in X_{mix}$, suppose there exists an $\epsilon > 0$ and a $0 < \delta < \frac{1}{2}$ such that $A^{i,j}_{x,\epsilon',\delta}$ and $A^{j,q}_{x,\epsilon'}$ have measure zero for all $\epsilon' \leq \epsilon$ and $j, q \neq i$, and the measure of $A^{i,j}_{x,\epsilon}$ is at least that of $A^{j,i}_{x,\epsilon}$.

Here the measure zero conditions are codifying the ideas that the point $x$ falls closer to $X_i$ than any other class on every line segment that intersects it, and there are no line segments between non-$i$ classes that intersect $x$. The condition that the measure of $A^{i,j}_{x,\epsilon}$ is at least that of $A^{j,i}_{x,\epsilon}$ handles asymmetric mixing distributions that concentrate on pathological values of $\lambda$. A visualization of Assumption 3.1 is provided in Section B of the Appendix. Now we can prove:

**Theorem 3.2.** Consider $k$-class classification where the supports $X_1, ..., X_k$ are finite and $\mathbb{P}_X$ corresponds to the discrete uniform distribution. If a point $x$ satisfies Assumption 3.1 with respect to a class $i$, then for every $h \in \mathrm{argmin}_{g \in \mathcal{C}^*} J_{mix}(g, \mathbb{P}_X, \mathbb{P}_f)$, we have that $h$ classifies $x$ as class $i$ and that $h$ is continuous at $x$.

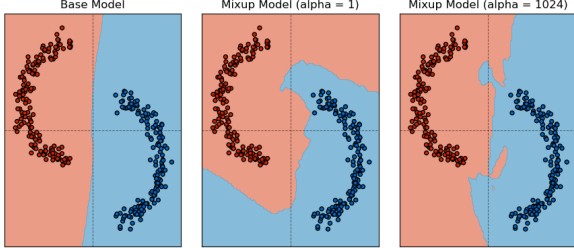

Figure 3: Decision boundary plots for standard and Mixup training on the two moons dataset of Pezeshki et al. (2020) with a class separation of 0.5. Each boundary represents the average of 10 training runs of 1500 epochs.

**Proof Sketch.** The limit in Lemma 2.3 can be shown to exist using the Lebesgue differentiation theorem, and we can bound the limit below since the $A_{x,\epsilon',\delta}^{i,j}$ have measure zero.

Assumption 2.9 implies Assumption 3.1 with respect to each class, and hence we get Theorem 2.10 as a corollary of Theorem 3.2 as mentioned in Section 2. To use Theorem 3.2 to understand generalization, we make the observation that a point $x$ can satisfy Assumption 3.1 while being a distance of up to $\frac{\min_j d(X_i, X_j)}{2}$ from some class $i$. This distance can be significantly farther than, for example, the optimal linear separator in a linearly separable dataset.

**Experiments.** To illustrate that Mixup can lead to more separation between points than a linear decision boundary, we consider the two moons dataset (Buitinck et al., 2013), which consists of two classes of points supported on semicircles with added Gaussian noise. Our motivation for doing so comes from the work of Pezeshki et al. (2020), in which it was noted that neural network models trained on a separated version of the two moons dataset essentially learned a linear separator while ignoring the curvature of the class supports. While Pezeshki et al. (2020) introduced an explicit regularizer to encourage a nonlinear decision boundary, we expect due to Theorem 3.2 that Mixup training will achieve a similar result without any additional modifications.

To verify this empirically, we train a two-layer neural network with 500 hidden units with and without Mixup, to have a 1-to-1 comparison with the setting of Pezeshki et al. (2020). We use $\alpha = 1$ and $\alpha = 1024$ for Mixup to capture a wide band of mixing densities. The version of the two moons dataset we use is also identical to that of the one used in the experiments of Pezeshki et al. (2020), and we are grateful to the authors for releasing their code under the MIT license. We do full-batch training with all other training, implementation, and compute details remaining the same as the previous section. Results are shown in Figure 3.

Our results affirm the observations of Pezeshki et al. (2020) and previous work (des Combes et al., 2018) that neural network training dynamics may ignore salient features of the dataset; in this case the "Base Model" learns to differentiate the two classes essentially based on the $x$-coordinate alone. On the other hand, the models trained using Mixup have highly nonlinear decision boundaries. Further experiments for different class separations and values of $\alpha$ are included in Section F of the Appendix.

### 3.2 WHEN MIXUP TRAINING LEARNS THE SAME CLASSIFIER

The experiments and theory of the previous sections have shown how a Mixup classifier can differ significantly from one learned through standard training. In this subsection, we now consider the opposing question - when is the Mixup classifier the same as the one learned through standard training? Prior work (Archambault et al., 2019) has considered when Mixup training coincides with certain adversarial training, and our results complement this line of work. The motivation for our results comes from the fact that a practitioner need not spend compute on Mixup training in addition to standard training in settings where the performance will be provably the same.

We consider the case of binary classification using a linear model $\theta^\top x$ on high-dimensional Gaussian data, which is a setting that arises naturally when training using Gaussian kernels. Specifically, we consider the dataset $X$ to consist of $n$ points in $\mathbb{R}^d$ distributed according to $\mathcal{N}(0, I_d)$ with $d > n$ (to be made more precise shortly). We also consider the mixing distribution to be any symmetric distribution supported on $[0, 1]$ (thereby including as a special case $\text{Beta}(\alpha, \alpha)$). We let the labels of

points in $X$ be $\pm 1$ (so that the sign of $\theta^\top x$ is the classification), and use $X_1$ and $X_{-1}$ to denote the individual class points. We will show that in this setting, the optimal Mixup classifier is the same (up to rescaling of $\theta$) as the ERM classifier learned using gradient descent with high probability. To do so we need some additional definitions.

**Definition 3.3.** We say $\hat\theta$ is an interpolating solution, if there exists $k > 0$ such that

$$\hat\theta^\top x_i = -\hat\theta^\top z_j = k \quad \forall x_i \in X_1, \; \forall z_j \in X_{-1}.$$

**Definition 3.4.** The maximum margin solution $\tilde\theta$ is defined through:

$$\tilde\theta := \underset{\|\theta\|_2 = 1}{\operatorname{argmax}} \left\{ \min_{x_i \in X_1, z_j \in X_{-1}} \left\{ \theta^\top x_i, -\theta^\top z_j \right\} \right\}$$

When the maximum margin solution coincides with an interpolating solution for the dataset $X$ (i.e. all the points are support vectors), we have that Mixup training leads to learning the max margin solution (up to rescaling).

**Theorem 3.5.** If the maximum margin solution for $X$ is also an interpolating solution for $X$, then any $\theta$ that lies in the span of $X$ and minimizes the Mixup loss $J_{mix}$ for a symmetric mixing distribution $\mathbb{P}_f$ is a rescaling of the maximum margin solution.

**Proof Sketch.** It can be shown that $\theta$ is an interpolating solution using a combination of the strict convexity of $J_{mix}$ as a function of $\theta$ and the symmetry of the mixing distribution.

**Remark 3.6.** For every $\theta$, we can decompose it as $\theta = \theta_X + \theta_{X^\perp}$ where $\theta_X$ is the projection of $\theta$ onto the subspace spanned by $X$. By definition we have that $\theta_{X^\perp}$ is orthogonal to all possible mixings of points in $X$. Hence, $\theta_{X^\perp}$ does not affect the Mixup loss or the interpolating property, so for simplicity we may just assume $\theta$ lies in the span of $X$.

To characterize the conditions on $X$ under which the maximum margin solution interpolates the data, we use a key result of Muthukumar et al. (2020), restated below. Note that Muthukumar et al. (2020) actually provide more settings in their paper, but we constrain ourselves to the one stated below for simplicity.

**Lemma 3.7.** [Theorem 1 in Muthukumar et al. (2020), Rephrased] Assuming $d > 10n \ln n + n - 1$, then with probability at least $1 - 2/n$, the maximum margin solution for $X$ is also an interpolating solution.

To tie the optimal Mixup classifier back to the classifier learned through standard training, we appeal to the fact that minimizing the empirical cross-entropy of a linear model using gradient descent leads to learning the maximum margin solution on linearly separable data (Soudry et al., 2018; Ji & Telgarsky, 2018). From this we obtain the desired result of this subsection:

**Corollary 3.8.** Under the same conditions as Lemma 3.7, the optimal Mixup classifier has the same direction as the classifier learned through minimizing the empirical cross-entropy using gradient descent with high probability.

## 4 CONCLUSION

The main contribution of our work has been to provide a theoretical framework for analyzing how Mixup training can differ from empirical risk minimization. Our results characterize a practical failure case of Mixup, and also identify conditions under which Mixup can provably minimize the original risk. They also show in the sense of margin why the generalization of Mixup classifiers can be superior to those learned through empirical risk minimization, while again identifying model classes and datasets for which the generalization of a Mixup classifier is no different (with high probability). We also emphasize that the generality of our theoretical framework allows most of our results to hold for any continuous mixing distribution. Our hope is that the tools developed in this work will see applications in future works concerned with analyzing the relationship between benefits obtained from Mixup training and properties of the training data.

## 5 ETHICS STATEMENT

We do not anticipate any direct misuses of this work due to its theoretical nature. That being said, the failure case of Mixup discussed in Section 2 could serve as a way for an adversary to potentially exploit a model trained using Mixup to classify data incorrectly. However, as this requires knowledge of the mixing distribution and other hyperparameters of the model, we do not flag this as a significant concern - we would just like to point it out for completeness.

## 6 REPRODUCIBILITY STATEMENT

Full proofs for all results in the main body of the paper can be found in Sections C and E of the Appendix. All of the code used to generate the plots and experimental results in this paper can be found at: `https://github.com/2014mchidamb/Mixup-Data-Dependency`. We have tried our best to organize the code to be easy to use and extend. Detailed instructions for how to run each type of experiment are provided in the `README` file included in the GitHub repository.

## ACKNOWLEDGEMENTS

Rong Ge, Muthu Chidambaram, Xiang Wang, and Chenwei Wu are supported in part by NSF Award DMS-2031849, CCF-1704656, CCF-1845171 (CAREER), CCF-1934964 (Tripods), a Sloan Research Fellowship, and a Google Faculty Research Award. Muthu would like to thank Michael Lin for helpful discussions during the early stages of this project.

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

## A   REVIEW OF DEFINITIONS AND ASSUMPTIONS

For convenience, we first recall the definitions and assumptions stated throughout the paper below.

$$\ell_{mix}(g, s, t, \lambda) = \begin{cases} -\log g^i(z_{st}) & s, t \in X_i \\ -\left(\lambda \log g^i(z_{st}) + (1 - \lambda) \log g^j(z_{st})\right) & s \in X_i, t \in X_j \end{cases}$$

$$J_{mix}^{i,j}(g, \mathbb{P}_X, \mathbb{P}_f) = \int_{X_i \times X_j \times [0,1]} \ell_{mix}(g, s, t, \lambda) \, d(\mathbb{P}_X \times \mathbb{P}_X \times \mathbb{P}_f)(s, t, \lambda)$$

$$J_{mix}(g, \mathbb{P}_X, \mathbb{P}_f) = \sum_{i=1}^{k} \sum_{j=1}^{k} J_{mix}^{i,j}(g, \mathbb{P}_X, \mathbb{P}_f)$$

$$A_{x,\epsilon}^{i,j} = \{(s, t, \lambda) \in X_i \times X_j \times [0, 1] : \lambda s + (1 - \lambda)t \in B_\epsilon(x)\}$$

$$A_{x,\epsilon,\delta}^{i,j} = \{(s, t, \lambda) \in X_i \times X_j \times [0, 1 - \delta] : \lambda s + (1 - \lambda)t \in B_\epsilon(x)\}$$

$$X_{mix} = \left\{ x \in \mathbb{R}^n : \bigcup_{i,j} A_{x,\epsilon}^{i,j} \text{ has positive measure for every } \epsilon > 0 \right\}$$

$$\xi_{x,\epsilon}^{i,j} = \int_{A_{x,\epsilon}^{i,j}} d(\mathbb{P}_X \times \mathbb{P}_X \times \mathbb{P}_f)(s, t, \lambda)$$

$$\xi_{x,\epsilon,\lambda}^{i,j} = \int_{A_{x,\epsilon}^{i,j}} \lambda \, d(\mathbb{P}_X \times \mathbb{P}_X \times \mathbb{P}_f)(s, t, \lambda)$$

**Definition 2.1.** Let $\mathcal{C}^*$ to be the subset of $\mathcal{C}$ for which every $h \in \mathcal{C}^*$ satisfies $h(x) = \lim_{\epsilon \to 0} \operatorname{argmin}_{\theta \in [0,1]^k} J_{mix}(\theta)|_{B_\epsilon(x)}$ for all $x \in X_{mix}$ when the limit exists. Here $J_{mix}(\theta)|_{B_\epsilon(x)}$ represents the Mixup loss for a constant function with value $\theta$ with the restriction of each term in $J_{mix}$ to the set $A_{x,\epsilon}^{i,j}$.

**Definition 2.6.** [3-Point Alternating Line] We define $\mathcal{X}_3^2$ to be the binary classification dataset consisting of the points $\{0, 1, 2\}$ classified as $\{1, 2, 1\}$. In our setup, this corresponds to $X_1 = \{0, 2\}$ and $X_2 = \{1\}$ with $\mathbb{P}_X = \frac{1}{3} 1_{\{0,1,2\}}$.

**Assumption 2.9.** For any point $x \in X_i$, there do not exist $u \in X$ and $v \in X_j$ for $j \neq i$ such that there is a $\lambda > 0$ for which $x = \lambda u + (1 - \lambda)v$.

**Assumption 3.1.** For a class $i$ and a point $x \in X_{mix}$, suppose there exists an $\epsilon > 0$ and a $0 < \delta < \frac{1}{2}$ such that $A_{x,\epsilon',\delta}^{i,j}$ and $A_{x,\epsilon'}^{j,q}$ have measure zero for all $\epsilon' \leq \epsilon$ and $j, q \neq i$, and the measure of $A_{x,\epsilon}^{i,j}$ is at least that of $A_{x,\epsilon}^{j,i}$.

**Definition 3.3.** We say $\hat{\theta}$ is an interpolating solution, if there exists $k > 0$ such that

$$\hat{\theta}^\top x_i = -\hat{\theta}^\top z_j = k \quad \forall x_i \in X_1, \, \forall z_j \in X_{-1}.$$

**Definition 3.4.** The maximum margin solution $\tilde{\theta}$ is defined through:

$$\tilde{\theta} := \operatorname*{argmax}_{\|\theta\|_2 = 1} \left\{ \min_{x_i \in X_1, z_j \in X_{-1}} \left\{ \theta^\top x_i, -\theta^\top z_j \right\} \right\}$$

## B   VISUALIZATIONS OF DEFINITIONS AND ASSUMPTIONS

Due to the technical nature of the definitions and assumptions above, we provide several visualizations in Figures 4 to 7 to help aid the reader's intuition for our main results.

## C   FULL PROOFS FOR SECTION 2

We now prove all results found in Section 2 of the main body of the paper in the order that they appear.

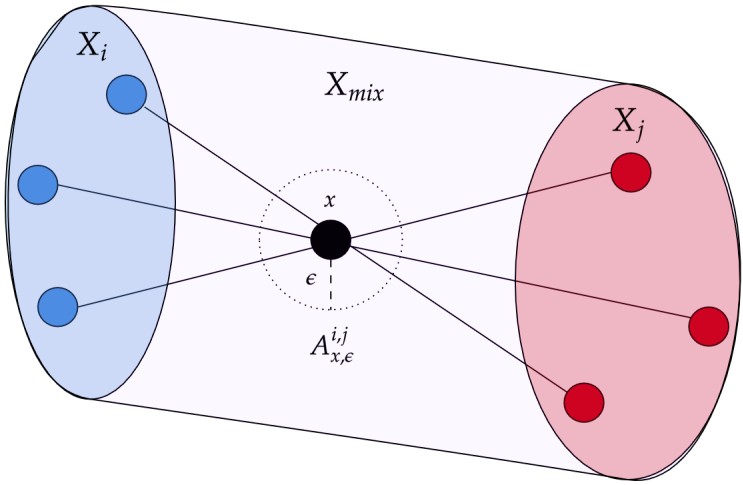

Figure 4: A visualization of $X_{mix}$ and $A_{x,\epsilon}^{i,j}$.

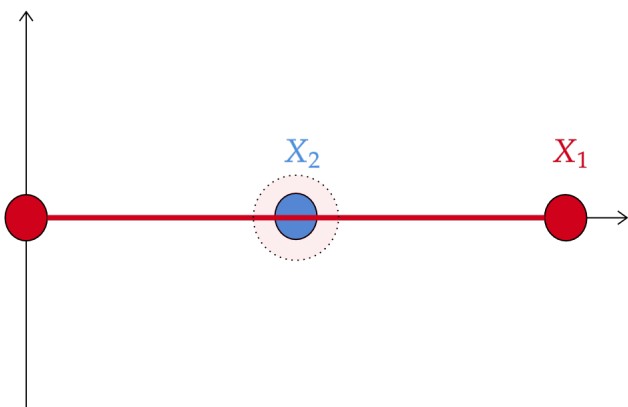

Figure 5: A visualization of the $\mathcal{X}_3^2$ dataset and how the Mixup sandwiching works.

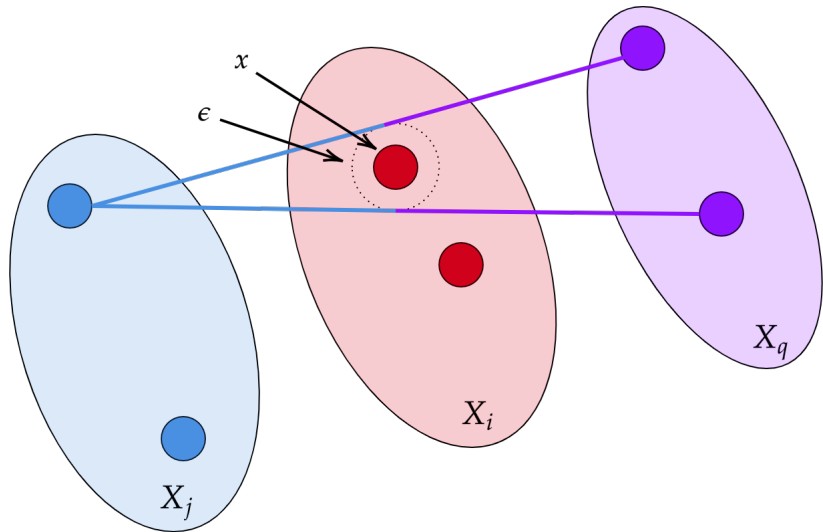

Figure 6: A visualization of Assumption 2.9, i.e. the "no collinearity" assumption.

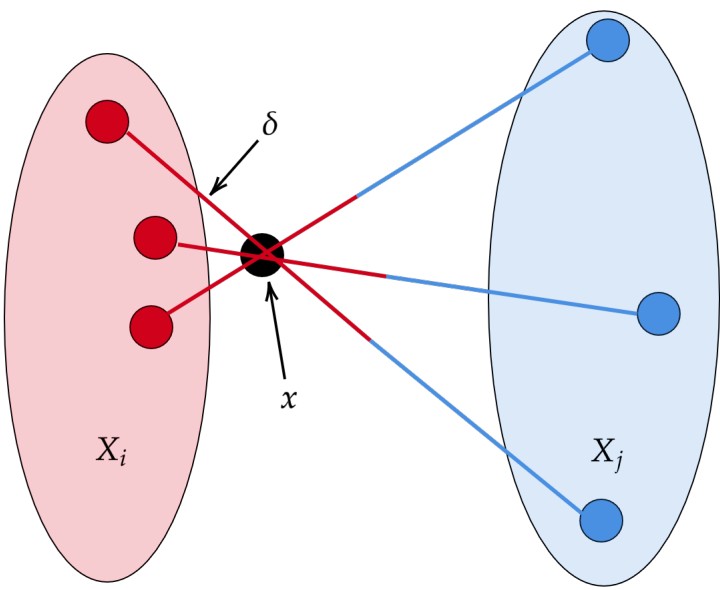

Figure 7: A visualization of Assumption 3.1. Once again, the key idea is that $x$ falls at most $\delta$ away from $X_i$ on every line between $X_i$ and $X_j$ that intersects it.

C.1 PROOFS FOR PROPOSITIONS, LEMMAS, AND THEOREMS 2.2 - 2.10

**Proposition 2.2.** Any function $h \in \operatorname{argmin}_{g \in \mathcal{C}^*} J_{mix}(g, \mathbb{P}_X, \mathbb{P}_f)$ satisfies $J_{mix}(h) \le J_{mix}(g)$ for any continuous $g \in \mathcal{C}$.

*Proof.* Intuitively, the idea is that when $h$ and $g$ differ at a point $x \in X_{mix}$, there must be a neighborhood of $x$ for which the constant function that takes value $h(x)$ has lower loss than $g$ due to the continuity constraint on $g$. We formalize this below.

Let $h$ be an arbitrary function in $\operatorname{argmin}_{g \in \mathcal{C}^*} J_{mix}(g, \mathbb{P}_X, \mathbb{P}_f)$ and let $g$ be a continuous function in $\mathcal{C}$. Consider a point $x \in X_{mix}$ such that the limit in Definition 2.1 exists and that $h(x) \ne g(x)$ (if such an $x$ did not exist, we would be done). Now let $\theta_h$ and $\theta_g$ be the constant functions whose values are $h(x)$ and $g(x)$ (respectively) on all of $\mathbb{R}^n$, and further let $\theta_{h_\delta} = \operatorname{argmin}_{\theta \in [0,1]^k} J_{mix}(\theta)|_{B_\delta(x)}$ (this is shown to be a single value in the proof of Lemma 2.3 below). Finally, as a convenient abuse of notation, we will use $J_{mix}(\epsilon')|_{B_\delta(x)}$ to indicate the result of replacing all $\log g^i$ terms in the integrands of $J_{mix}(g)|_{B_\delta(x)}$ with $\epsilon'$, as shown below (note that in doing so, we can combine the $\lambda$ and $1 - \lambda$ terms from mixing classes $i$ and $j$; this simplifies the $J_{mix}^{i,j}$ expression obtained in the proof of Lemma 2.3).

$$J_{mix}(\epsilon')|_{B_\delta(x)} = -\epsilon' \sum_{i=1}^{k} \sum_{j=1}^{k} \xi_{x,\epsilon}^{i,j}$$

Since $\theta_h \ne \theta_g$, we have that there exists a $\delta' > 0$ such that for $\delta \le \delta'$ we have $|\theta_{h_\delta} - \theta_g| = \epsilon > 0$. From this we get that there exists $\epsilon' > 0$ depending only on $\epsilon$ and $\theta_g$ such that:

$$J_{mix}(\theta_g)|_{B_\delta(x)} - J_{mix}(\theta_h)|_{B_\delta(x)} \ge J_{mix}(\epsilon')|_{B_\delta(x)}$$

Now by the continuity of $g$ (and thus the continuity of $\log g^i$), we may choose $\delta \le \delta'$ such that $J_{mix}(g)|_{B_\delta(x)} \in J_{mix}(\theta_g)|_{B_\delta(x)} \pm J_{mix}(\epsilon')|_{B_\delta(x)}$. This implies $J_{mix}(g)|_{B_\delta(x)} \ge J_{mix}(\theta_h)|_{B_\delta(x)}$, and since $x$ was arbitrary (within the initially mentioned constraints, as outside of them $h$ is unconstrained) we have the desired result. $\square$

**Lemma 2.3.** For any point $x \in X_{mix}$ and $\epsilon > 0$, there exists a continuous function $h_\epsilon$ satisfying:

$$h_\epsilon^i(x) = \frac{\xi_{x,\epsilon}^{i,i} + \sum_{j \ne i} \left( \xi_{x,\epsilon,\lambda}^{i,j} + (\xi_{x,\epsilon}^{j,i} - \xi_{x,\epsilon,\lambda}^{j,i}) \right)}{\sum_{q=1}^{k} \left( \xi_{x,\epsilon}^{q,q} + \sum_{j \ne q} \left( \xi_{x,\epsilon,\lambda}^{q,j} + (\xi_{x,\epsilon}^{j,q} - \xi_{x,\epsilon,\lambda}^{j,q}) \right) \right)} \tag{1}$$

With the property that $\lim_{\epsilon \to 0} h_\epsilon(x) = h(x)$ for every $h \in \operatorname{argmin}_{g \in \mathcal{C}^*} J_{mix}(g, \mathbb{P}_X, \mathbb{P}_f)$ when the limit exists.

*Proof.* Firstly, the condition that $x \in X_{mix}$ is necessary, since if $\bigcup_{i,j} A_{x,\epsilon}^{i,j}$ has measure zero the LHS of Equation 1 is not even defined.

Now we simply take $h_\epsilon(x) = \operatorname{argmin}_{\theta \in [0,1]^k} J_{mix}(\theta)|_{B_\epsilon(x)}$ as in Definition 2.1 and show that $h_\epsilon$ is well-defined.

Since the argmin is over constant functions, we may unpack the definition of $J_{mix}|_{B_\epsilon(x)}$ and pull each of the $\log \theta^i(z_{st}(\lambda))$ terms out of the integrands and rewrite them simply as $\log \theta^i$. Doing so, we obtain:

$$J_{mix}^{i,j} = -\left( \xi_{x,\epsilon,\lambda}^{i,j} \log \theta^i + (\xi_{x,\epsilon}^{i,j} - \xi_{x,\epsilon,\lambda}^{i,j}) \log \theta^j \right)$$

Plugging the above back into $J_{mix}$ and collecting the $\log \theta^i$ terms as $i = 1, ..., k$ we get:

$$J_{mix}(g, \mathbb{P}_X, \mathbb{P}_f)|_{B_\epsilon(x)} = -\sum_{i=1}^{k} \left( \xi_{x,\epsilon}^{i,i} + \sum_{j \ne i} \left( \xi_{x,\epsilon,\lambda}^{i,j} + (\xi_{x,\epsilon}^{j,i} - \xi_{x,\epsilon,\lambda}^{j,i}) \right) \right) \log \theta^i$$

Where the first part of the summation above is from mixing $X_i$ with itself, and the second part of the summation corresponds to the $\lambda$ and $(1-\lambda)$ components of mixing $X_i$ with $X_j$. Discarding the terms for which the coefficients above are 0 (the associated $\theta^i$ terms are taken to be 0, as anything else is suboptimal due to the summation constraint), we are left with a linear combination of $-\log\theta^{i_1}, -\log\theta^{i_2}, ..., -\log\theta^{i_m}$, where the set $\{i_1, i_2, ..., i_m\}$ is a subset of $\{1, ..., k\}$. If the aforementioned set consists of only a single index $i$, then the unique optimizer is of course $\theta^i = 1$. On the other hand, if there are multiple $\theta^{i_p}$, we cannot minimize $J_{mix}|_{B_\epsilon(x)}$ by choosing any of the $\theta^{i_p}$ to be 0, and as a consequence we also cannot choose any of the $\theta^{i_p}$ to be 1 since they are constrained to add to 1.

As such, we may consider each of the $\theta^{i_p} \in [\delta, 1-\delta]$ for some $\delta > 0$. With this consideration in mind, $J_{mix}|_{B_\epsilon(x)}$ is strictly convex in terms of the $\theta^{i_p}$, since the Hessian of a linear combination of $-\log\theta^{i_p}$ will be diagonal with positive entries when the arguments of the log terms are strictly greater than 0 and less than 1. Thus, as $[\delta, 1-\delta]^m$ is compact, there exists a unique solution to $h_\epsilon(x) = \operatorname{argmin}_{\theta \in [0,1]^k} J_{mix}(\theta)|_{B_\epsilon(x)}$, justifying the use of equality. This unique solution is easily computed via Lagrange multipliers, and the solution is given in Equation 1.

We have thus far defined $h_\epsilon$ on $X_{mix}$, and it remains to check that this construction of $h_\epsilon$ corresponds to the restriction of a continuous function from $\mathbb{R}^n \to [0,1]^k$. To do so, we first note that $X_{mix}$ is closed, since any limit point $x'$ of $X_{mix}$ must necessarily either be contained in one of the supports $X_i$ (due to compactness) or on a line segment between/within two supports (else it has positive distance from $X_{mix}$), and every $\epsilon'$-neighborhood of $x'$ contains points $x$ for which $\bigcup_{i,j} A_{x,\epsilon}^{i,j}$ has positive measure for every $\epsilon$ (immediately implying that $\bigcup_{i,j} A_{x',\epsilon'}^{i,j}$ has positive measure).

Now we can check that $h_\epsilon$ is continuous on $X_{mix}$ as follows. Consider a sequence $x_m \to x^*$; we wish to show that $h_\epsilon(x_m) \to h_\epsilon(x^*)$. Since the codomain of $h_\epsilon$ is compact, the sequence $h_\epsilon(x_m)$ must have a limit point $h^*$. Furthermore, since each $h_\epsilon(x_m)$ is the unique minimizer of $J_{mix}|_{B_\epsilon(x_m)}$, we must have that $h^*$ is the unique minimizer of $J_{mix}|_{B_\epsilon(x^*)}$, implying that $h^* = h_\epsilon(x^*)$. We have thus established the continuity of $h_\epsilon$ on $X_{mix}$, and since $X_{mix}$ is closed (in fact, compact), we have by the Tietze Extension Theorem that $h_\epsilon$ can be extended to a continuous function on all of $\mathbb{R}^n$.

Finally, if $\lim_{\epsilon \to 0} h_\epsilon(x)$ exists, then it is by definition $h(x)$ for any $h \in \mathcal{C}^*$ and therefore for any $h \in \operatorname{argmin}_{g \in \mathcal{C}^*} J_{mix}(g, \mathbb{P}_X, \mathbb{P}_f)$. $\qquad \square$

**Corollary C.1** (Symmetric Version). If $\mathbb{P}_f$ is symmetric, then we may simplify $h_\epsilon^i$ to be:

$$h_\epsilon^i(x) = \frac{\xi_{x,\epsilon}^{i,i} + 2\sum_{j \neq i} \xi_{x,\epsilon,\lambda}^{i,j}}{\sum_{q=1}^k \left(\xi_{x,\epsilon}^{q,q} + 2\sum_{j \neq q} \xi_{x,\epsilon,\lambda}^{q,j}\right)}$$

*Proof.* In the symmetric case, we have $\xi_{x,\epsilon,\lambda}^{i,j} = \xi_{x,\epsilon}^{j,i} - \xi_{x,\epsilon,\lambda}^{j,i}$. $\qquad \square$

**Proposition 2.5.** Let $X_1 = \{(0,1), (0,-1)\}$ and let $X_2 = \{(1,0), (-1,0)\}$, with $\mathbb{P}_X$ being discrete uniform over $X_1 \cup X_2$ and $\mathbb{P}_f$ being continuous uniform over $[0,1]$. Then the Mixup-optimal classifier $h$ is discontinuous at $(0,0)$.

*Proof.* Considering the point $x = (0,0)$, we note that only $\xi_{x,\epsilon}^{1,1}$ and $\xi_{x,\epsilon}^{2,2}$ are non-zero for all $\epsilon > 0$. Furthermore, we have that $\xi_{x,\epsilon}^{1,1} = \xi_{x,\epsilon}^{2,2} = \frac{\epsilon}{4}$ since the $\mathbb{P}_f$-measure of $B_\epsilon(\frac{1}{2})$ is $2\epsilon$ and the sets $A_{x,\epsilon}^{1,1}, A_{x,\epsilon}^{2,2}$ have $\mathbb{P}_X$-measure $\frac{1}{8}$. From this we have that the limit $\lim_{\epsilon \to 0} h_\epsilon(x) = \begin{bmatrix} \frac{1}{2} & \frac{1}{2} \end{bmatrix}^\top$ is the Mixup-optimal value at $x$.

On the other hand, every other point on the line segment connecting the points in $X_1$ will have an $\epsilon$-neighborhood disjoint from the line segment between the points in $X_2$ (and vice versa), so we will have $h_{\epsilon'}^1(x) = 1$ for all $\epsilon' \leq \epsilon$. This implies that $h_{\epsilon'}(x) \to [1 \; 0]$ (and an identical result for points on the $X_2$ line segment), so we have that the Mixup-optimal classifier is discontinuous at $(0,0)$ as desired. $\qquad \square$

**Theorem 2.7.** Let $\mathbb{P}_f$ have associated density $\text{Beta}(\alpha, \alpha)$. Then for any classifier $h_\epsilon$ on $\mathcal{X}_3^2$ (as defined in Lemma 2.3), we may choose $\alpha$ such that $h_\epsilon$ does not achieve 0 classification error on $\mathcal{X}_3^2$.

*Proof.* Fix a classifier $h_\epsilon$ as defined in Lemma 2.3 on $\mathcal{X}_3^2$. Now for $Y \sim \text{Beta}(\alpha, \alpha)$ we have by the fact that $\text{Beta}(\alpha, \alpha)$ is strictly subgaussian that $P\left(\left|Y - \frac{1}{2}\right| \leq \epsilon\right) \geq 1 - 2\exp\left(-\epsilon^2/(2\sigma^2)\right)$ where $\sigma^2 = \frac{1}{4\alpha+2}$ is the variance of $\text{Beta}(\alpha, \alpha)$. As a result, we can choose $\alpha > \frac{1}{2}\left(\frac{\log 4}{\epsilon^2} - 1\right)$ to guarantee that $P\left(\left|Y - \frac{1}{2}\right| \leq \epsilon\right) > \frac{1}{2}$ and therefore that $\xi_{1,\epsilon}^{1,1} > \frac{1}{9} = \xi_{1,\epsilon}^{2,2}$.

Now we have by Lemma 2.3 (or more precisely, Corollary C.1) that:

$$h_\epsilon^1(1) = \frac{\xi_{1,\epsilon}^{1,1} + 2\xi_{1,\epsilon,\lambda}^{1,2}}{\xi_{1,\epsilon}^{1,1} + 4\xi_{1,\epsilon,\lambda}^{1,2} + \xi_{1,\epsilon}^{2,2}} > \frac{1}{2}$$

Thus, we have shown that $h_\epsilon$ will classify the point 1 as class 1 despite it belonging to class 2. $\square$

**Proposition 2.8.** Consider $k$-class classification where the supports $X_1, ..., X_k$ are finite and $\mathbb{P}_X$ corresponds to the discrete uniform distribution. Then for every $h \in \text{argmin}_{g \in \mathcal{C}^*} J_{mix}(g, \mathbb{P}_X, \mathbb{P}_f)$, we have that $h^i(x) = 1$ on $X_i$.

*Proof.* The full proof is not much more than the proof sketch in the main body. For $x \in X_i$, we have that $\xi_{x,\epsilon}^{j,q} \to 0$ and $\xi_{x,\epsilon,\lambda}^{j,q} \to 0$ as $\epsilon \to 0$ for every $(j, q) \neq (i, i)$, while $\xi_{x,\epsilon}^{i,i} \to \frac{1}{\left|\bigcup_i X_i\right|^2}$ and $\xi_{x,\epsilon,\lambda}^{i,i} \to \frac{\mathbb{E}_{\mathbb{P}_f}[\lambda]}{\left|\bigcup_i X_i\right|^2}$. As a result, we have $\lim_{\epsilon \to 0} h_\epsilon^i(x) = 1$ as desired. $\square$

**Theorem 2.10.** We consider the same setting as Proposition 2.8 and further suppose that Assumption 2.9 is satisfied. Then for every $h \in \text{argmin}_{g \in \mathcal{C}^*} J_{mix}(g, \mathbb{P}_X, \mathbb{P}_f)$, we have that $h^i(x) = 1$ on $X_i$ and that $h$ is continuous on $X$.

*Proof.* Obtained as a corollary of Theorem 3.2. $\square$

## C.2 PROOF FOR THEOREM 2.11

Prior to proving Theorem 2.11, we first introduce some additional notation as well as a lemma that will be necessary for the proof.

**Notation:** Throughout the following $m$ corresponds to the number of data points being considered (as in Theorem 2.11), and as a shorthand we use $[m]$ to indicate $\{1, ..., m\}$. Additionally, for two matrices $A$ and $B$, we use the notation $[A, B]$ to indicate the matrix formed by the concatenation of the columns of $B$ to $A$. We use $e_i$ to denote the $i$-th basis vector in $R^m$, and use $e_i'$ to denote the $i$-th basis vector in $R^{\binom{m}{2}}$. Let $A \in \mathbb{R}^{\binom{m}{2} \times m}$ be the "Mixup matrix" where each row has two 1s and the other entries are 0, representing a mixture of the two data points whose associated indices have a 1. The $\binom{m}{2}$ rows enumerate all the possible mixings of the $m$ data points. In this way, $A$ is uniquely defined up to a permutation of rows. We can pick any representative as our $A$ matrix, and prove the following lemma.

**Lemma C.2.** Assume $m > 6$, and $P \in \mathbb{R}^{\binom{m}{2} \times \binom{m}{2}}$ is a permutation matrix. If $PA$ is not a permutation of the columns of $A$, then the rank of $[A, PA]$ is larger than $m$.

Using this lemma we can prove Theorem 2.11.

**Theorem 2.11.** Suppose $\{x_1, ..., x_m\}$ with $m \geq 6$ are sampled from $X$ according to $\mathbb{P}_X$, and that $\mathbb{P}_X$ has a density. Then with probability 1, we can uniquely determine the points $\{x_1, ..., x_m\}$ given only the $\binom{m}{2}$ midpoints $\{x_{i,j}\}_{1 \leq i < j \leq m}$.

*Proof.* We only need to show that the set of samples that cannot be uniquely determined from their midpoints has Lebesgue measure zero, since $\mathbb{P}_X$ is absolutely continuous with respect to the Lebesgue measure. It suffices to show this for the first entry (dimension) of the $x_i$'s, as the result then follows for all dimensions. Let $\{x_i'\}$ be another sample of $m$ points. For convenience, we group the first entries of the data points $\{x_i\}_{i=1}^m$ into a vector $w^* \in \mathbb{R}^m$, and similarly obtain $w \in \mathbb{R}^m$ from $\{x_i'\}_{i=1}^m$. Suppose $w^* \in \mathbb{R}^m$ is not a permutation of $w \in \mathbb{R}^m$ but that they have the same set of Mixup points. We only need to show that the set of such $w^*$ has measure zero.

Suppose $A$ is a Mixup matrix and $P$ is a permutation matrix. Suppose $PA$ is not a permutation of the columns in $A$. We would need $Aw^* = PAw$, which is equivalent to $[A, PA][(w^*)^\top, -w^\top]^\top = 0$ (here $[(w^*)^\top, -w^\top]^\top$ indicates the vector resulting from concatenating $-w$ to $w^*$). According to Lemma C.2, we know the rank of $[A, PA]$ is at least $m+1$, which implies that the solution set of $w^*$ is at most $m-1$.

So fixing $A$ and $P$, the set of non-recoverable $w^*$ has measure zero. There are only a finite number of combinations of $A$ and $P$. Thus, considering all of these $A$ and $PA$, the full set of non-recoverable $w^*$ still has measure zero. $\qquad\square$

### C.2.1   PROOF OF SUPPORTING LEMMA

**Lemma C.2.** Assume $m > 6$, and $P \in \mathbb{R}^{\binom{m}{2} \times \binom{m}{2}}$ is a permutation matrix. If $PA$ is not a permutation of the columns of $A$, then the rank of $[A, PA]$ is larger than $m$.

*Proof.* First, we show that both the ranks of $A$ and $PA$ are $m$. For all $i \in [m-1]$, define $u_i = e_1 + e_{i+1}$, and define $u_m = e_2 + e_3$. Note that these $m$ vectors are all rows of $A$. The first $(m-1)$ vectors $\{u_i\}_{i=1}^{m-1}$ are linearly independent because each $u_i$ has a unique direction $e_{i+1}$ that is not a linear combination of any other vectors in $\{u_i\}_{i=1}^{m-1}$. Besides, we know that the span of $\{u_i\}_{i=1}^{m-1}$ is a subspace of $\{v \in \mathbb{R}^m : \sum_{i=2}^{m} v(i) = v(1)\}$ where $v(i)$ is the $i$-th entry of $v$. Therefore, $u_m$ doesn't lie in the span of $\{u_i\}_{i=1}^{m-1}$, which implies that these $n$ vectors $\{u_i\}_{i=1}^{n}$ are linearly independent. This shows that $A$ is at least rank $m$. Since $A$ only has $m$ columns, we know that the rank of $A$ is $m$. The matrix $PA$ is also rank $m$ because $P$ is full rank.

Therefore, to show that the rank of $[A, PA]$ is larger than $m$, we only need to find a vector $v$ that lies in the column span of $A$ but is not in the column span of $PA$. To do this, we need the following claim:

**Claim 1.** There exists a row index subset $I \subseteq [\binom{m}{2}]$ with size $(m-1)$ such that the sub-matrix composed by the rows of $A$ with indices in $I$ has a column that is all-one vector, but every column of the sub-matrix composed by the rows of $PA$ with indices in $I$ is not all-one vector.

**Proof of Claim 1.** By the definition of $A$, each column of $A$ has only $(m-1)$ ones and the other entries are zero. Therefore, the position of $(m-1)$ ones in a column of $A$ can uniquely determine that column vector. Besides, for an index set $I$ with size $(m-1)$, the sub-matrix composed by the rows of $A$ with indices in $I$ cannot have two columns that are both all-one vector. This is because otherwise $A$ will have duplicate rows, which contradicts the definition of $A$. Therefore, there are $m$ possible choices of $I$, each of which corresponds to the positions of all ones in a column of $A$.

Assume by contradiction that for these $m$ choices of $I$, there exist a column of the sub-matrix composed by the rows of $PA$ with indices in $I$ that is all-one vector. Then these $m$ choices of $I$ also correspond to the positions of all ones in a column of $PA$. This means that the columns of $A$ and $PA$ are the same up to permutations, which contradicts the assumption of our theorem. This finishes the proof of this claim.

Now define $B_1$ as the sub-matrix composed by the rows of $A$ with indices in $I$, and $C_1$ as the sub-matrix composed by the rows of $PA$ with indices in $I$. Without loss of generality, suppose $I = [m-1]$, and suppose the first column of $B_1$ is all-one vector. Let $u = -e_1 + \sum_{i=2}^{m} e_i \in \mathbb{R}^m$, we know that $Au = 2 \sum_{i=m}^{\binom{m}{2}} e_i'$, i.e., the first $(m-1)$ entries of $Au$ are 0, and the other entries are 2. Define $v \triangleq Au$, we are going to show that $v$ is not in the column span of $PA$. Let $C_2$ be the sub-matrix in $PA$ consisting of the rows that are not in $C_1$, then the following claim shows that $C_2$ has full column rank.

**Claim 2.** $rank(C_2) = m$.

**Proof of Claim 2.** In this proof, we will consider each column of $C_2$ as a vertex. Since each row of $C_2$ has only two 1s, we view each row as an edge connecting the two columns which correspond to the two 1s in that row. From the definition of $A$ we know that the graph we constructed is a simple undirected graph. Then we are going to show that we can select $m$ "edges" from $C_2$ which are linearly independent.

There are $\binom{m}{2} - (m-1)$ edges in $C_2$. From $m > 6$ we know that $\binom{m}{2} - (m-1) > \frac{m^2}{4}$, so from Turán's theorem, $C_2$ contains at least one triangle. Assume this triangle is $(i, j, k)$, then we select edges $(i, j)$, $(j, k)$ and $(i, k)$ and define $E_3 = \{i, j, k\}$. $\forall r \in [m], r \geq 3$, we select an edge connecting $E_r$ with $[m] \setminus E_r$. Assume that edge is $(s, t)$ where $s \in E_r$ and $t \notin E_r$, we then add $t$ to $E_r$, i.e., $E_{r+1} = E_r \cup \{t\}$. In the next two paragraphs, we are going to show that there are always edges between $E_r$ and $[m] \setminus E_r$ (so we can successfully select $m$ edges in total), and the $m$ edges we selected are linearly independent.

In matrix $PA$, there are $r(m - r)$ edges between $E_r$ and $[m] \setminus E_r$. Since $C_2$ is constructed by deleting $(m - 1)$ edges from $PA$, the number of edges left between $E_r$ and $[m] \setminus E_r$ is at least $r(m - r) - (m - 1)$. When $3 \leq r \leq m - 2$, we have $r(m - r) - (m - 1) > 0$. When $r = m - 1$, the only case where there is no edge between $E_r$ and $[m] \setminus E_r$ is when all edges from vertex $[m] \setminus E_r$ is in $C_1$, which means that $C_1$ has a column that is all-one vector and is a contradiction. Therefore, there are always edges between $E_r$ and $[m] \setminus E_r$ and we can successfully select $m$ edges.

Then we only need to show that these $m$ selected edges are linearly independent. We use $\{u_i\}_{i=1}^m$ to denote the vectors that correspond to these edges, i.e., $u_1 = e_i + e_j$, $u_2 = e_j + e_k$, $u_3 = e_i + e_k$, $\cdots$ Assume by contradiction that they are linearly independent, then there exists $x \in \mathbb{R}^m$, $x \neq 0$ such that $\sum_{i=1}^m x(i)u_i = 0$. By the selection process of the edges, we know that $\forall r \geq 4$, $u_r$ has a unique direction, so $x(r) = 0$. Therefore, $x(1)u_1 + x(2)u_2 + x(3)u_3 = 0$. Since $\{u_1, u_2, u_3\}$ are linearly independent, we have $x = 0$, which is a contradiction. Thus, these $m$ selected edges are linearly independent, proving the claim.

Now assume by contradiction that $v$ is in the column span of $PA$, then $\exists w \in \mathbb{R}^m$ such that $v = PAw$. Let $v_2 \in \mathbb{R}^{\binom{m}{2}-(m-1)}$ be the bottom $\binom{m}{2} - (m-1)$ entries of $v$, then $v_2 = C_2 w$. Define $w_0$ to be the all-one vector in $\mathbb{R}^m$, we know that $w_0$ is a valid solution to $v_2 = C_2 w$. Since $C_2$ has full column rank, $w_0$ must be the unique solution to $v_2 = C_2 w$. This implies that $v = PAw_0$. However, we know that $PAw_0 = 2\sum_{i=1}^{\binom{m}{2}} e_i' \neq v$, which is a contradiction. Thus, $v$ is not in the column span of $A$, which finishes the proof of the lemma. $\square$

# D   ADDITIONAL EXPERIMENTS FOR SECTION 2

## D.1   EXPERIMENTS FOR SECTION 2.3

As noted in the main paper, it is not difficult to extend Definition 2.6 to construct datasets on which Mixup training empirically fails for small values of $\alpha$. The observation to make is that a major part of the proof of Theorem 2.7 is the fact that the same-point mixing probability of point 1 is small relative to the same-class mixing probability of class 0. As the former probability decreases quadratically with the dataset size $m$, we are motivated to consider extensions of $\mathcal{X}_3^2$ consisting of more points and more classes. Towards that end, we consider the following generalization of $\mathcal{X}_3^2$:

**Definition D.1** ($m$-Point $k$-Class Alternating Line). We define $\mathcal{X}_m^k$ to be the $k$-class classification dataset consisting of the points $\{0, 1, ..., m-1\}$ classified according to their value mod $k$ incremented by 1. As before, $\mathbb{P}_X$ is the normalized counting measure on $\mathcal{X}_m^k$.

We now consider training a two-layer feedforward network on $\mathcal{X}_{10}^2$ and $\mathcal{X}_{10}^{10}$ using the same procedure and hyperparameters as in the main paper. The results are shown in Figure 8 below.

Here we see that even at $\alpha = 1$, Mixup training fails to minimize the original empirical risk on $\mathcal{X}_{10}^2$ and $\mathcal{X}_{10}^{10}$, whereas in the main paper we noted that at $\alpha = 1$ Mixup training had no issues minimizing the original risk on $\mathcal{X}_3^2$. Interestingly, we find that even standard training does not completely minimize the original empirical risk on $\mathcal{X}_{10}^2$, once again perhaps a result of the regularity of the two-layer network model (although this is merely a hypothesis, and we do not analyze this phenomenon further here).

## D.2   EXPERIMENTS FOR SECTION 2.4

Here we include the additional experiments mentioned in Section 2.4, namely the results of training ResNet-18 on MNIST, CIFAR-10, and CIFAR-100 with and without Mixup for $\alpha = 1, 32$, and $128$ (with the other hyperparameters being as described in Section 2.4). The results are shown in Figure 9.

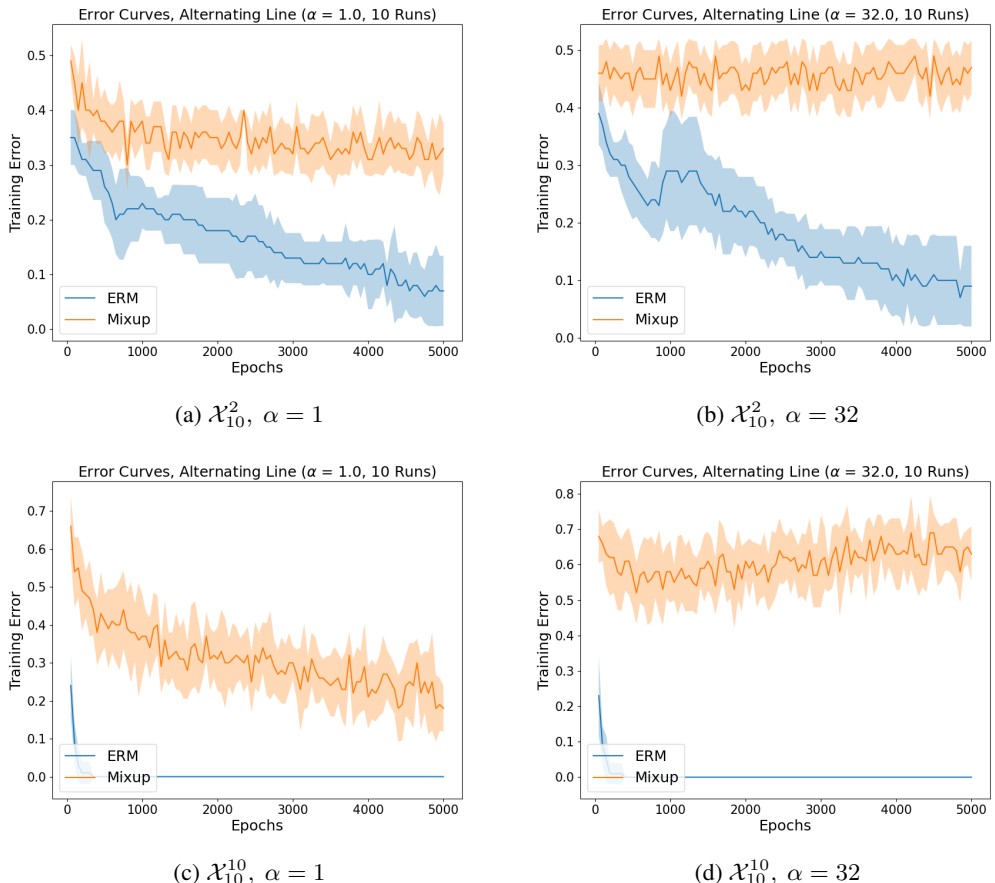

(a) $\mathcal{X}_{10}^2$, $\alpha = 1$          (b) $\mathcal{X}_{10}^2$, $\alpha = 32$

(c) $\mathcal{X}_{10}^{10}$, $\alpha = 1$          (d) $\mathcal{X}_{10}^{10}$, $\alpha = 32$

Figure 8: Training error plots for Mixup and regular training on $\mathcal{X}_{10}^2$ and $\mathcal{X}_{10}^{10}$. Each curve corresponds to the mean of 10 training runs, and the area around each curve represents a region of one standard deviation. Note that the ERM curves appear slightly different across $\alpha$ values due to changes in $y$-axis scale.

As mentioned in the main body, these choices of $\alpha$ only lead to Mixup performing more similarly to ERM than $\alpha = 1024$.

## E  FULL PROOFS FOR SECTION 3

### E.1  PROOF OF THEOREM 3.2

**Theorem 3.2.** Consider $k$-class classification where the supports $X_1, ..., X_k$ are finite and $\mathbb{P}_X$ corresponds to the discrete uniform distribution. If a point $x$ satisfies Assumption 3.1 with respect to a class $i$, then for every $h \in \text{argmin}_{g \in \mathcal{C}^*} J_{mix}(g, \mathbb{P}_X, \mathbb{P}_f)$, we have that $h$ classifies $x$ as class $i$ and that $h$ is continuous at $x$.

*Proof.* For two points $p, q$ in the supports $X_1, ..., X_k$ with a line segment between them intersecting $x$, let $\lambda(p, q, x)$ denote the value of $\lambda$ for which $\lambda p + (1 - \lambda)q = x$. Since the supports are finite, we have that there exists $\epsilon_1 > 0$ such that for all $\epsilon \leq \epsilon_1$ we have that:

$$\xi_{x,\epsilon}^{i,j} = \sum_p \sum_q \int_{B_\epsilon(\lambda(p,q,x))} d\mathbb{P}_f$$

Where the summations are over all points $p, q$ with line segments containing $x$. Now we have by the Lebesgue differentiation theorem applied to the integral term in the summations above (this

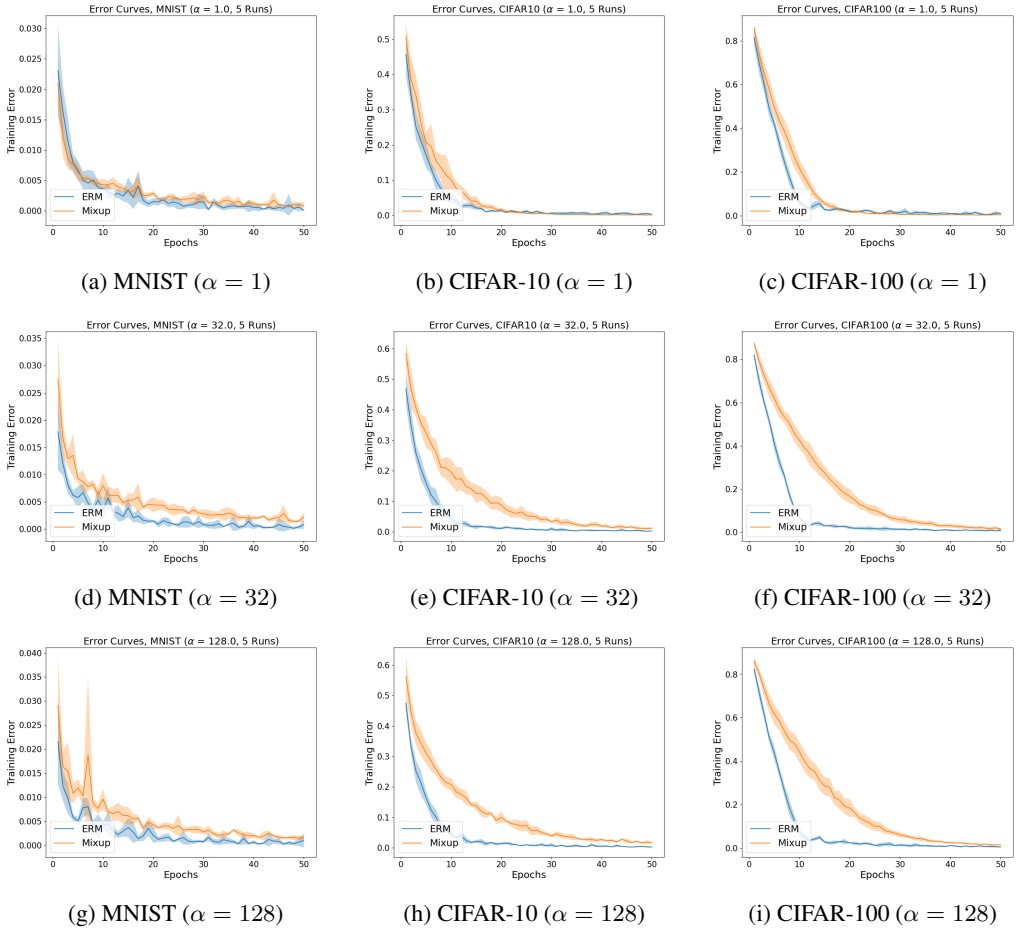

Figure 9: Mean and single standard deviation of 5 training runs for Mixup ($\alpha = 1, 32, 128$) and ERM on the *original training data*.

is possible because $\mathbb{P}_f$ has a density) that the following limit (as well as each limit for the other coordinate functions) exists:

$$
\lim_{\epsilon \to 0} h_\epsilon^i = \lim_{\epsilon \to 0} \frac{\xi_{x,\epsilon}^{i,i} + \sum_{j \neq i} \left( \xi_{x,\epsilon,\lambda}^{i,j} + (\xi_{x,\epsilon}^{j,i} - \xi_{x,\epsilon,\lambda}^{j,i}) \right)}{\sum_{q=1}^{k} \left( \xi_{x,\epsilon}^{q,q} + \sum_{j \neq q} \left( \xi_{x,\epsilon,\lambda}^{q,j} + (\xi_{x,\epsilon}^{j,q} - \xi_{x,\epsilon,\lambda}^{j,q}) \right) \right)}
$$

$$
= \lim_{\epsilon \to 0} \frac{\frac{1}{\epsilon}\xi_{x,\epsilon}^{i,i} + \frac{1}{\epsilon}\sum_{j \neq i} \left( \xi_{x,\epsilon,\lambda}^{i,j} + (\xi_{x,\epsilon}^{j,i} - \xi_{x,\epsilon,\lambda}^{j,i}) \right)}{\frac{1}{\epsilon}\sum_{q=1}^{k} \left( \xi_{x,\epsilon}^{q,q} + \sum_{j \neq q} \left( \xi_{x,\epsilon,\lambda}^{q,j} + (\xi_{x,\epsilon}^{j,q} - \xi_{x,\epsilon,\lambda}^{j,q}) \right) \right)}
$$

Now by Assumption 3.1, we need only consider the $\xi_{x,\epsilon}^{i,i}, \xi_{x,\epsilon}^{i,j}, \xi_{x,\epsilon,\lambda}^{i,j}$ terms (as there are no other line segments that contain $x$). Furthermore, we have that there exists $\epsilon_2 > 0$ such that for all $\epsilon \leq \epsilon_2$ we have $\lambda(p, q, x) - \epsilon \geq \frac{1}{2}$ for $p \in X_i$ and $q \in X_j$, and that the measure of $A_{x,\epsilon}^{i,j}$ is at least that of $A_{x,\epsilon}^{j,i}$. From this we get that for all $\epsilon \leq \min(\epsilon_1, \epsilon_2)$:

$$
h_\epsilon^i = \frac{\xi_{x,\epsilon}^{i,i} + \sum_{j \neq i} \left( \xi_{x,\epsilon,\lambda}^{i,j} + (\xi_{x,\epsilon}^{j,i} - \xi_{x,\epsilon,\lambda}^{j,i}) \right)}{\xi_{x,\epsilon}^{i,i} + \sum_{j \neq i} \left( \xi_{x,\epsilon,\lambda}^{i,j} + \xi_{x,\epsilon,\lambda}^{j,i} + (\xi_{x,\epsilon}^{j,i} - \xi_{x,\epsilon,\lambda}^{j,i}) + (\xi_{x,\epsilon}^{i,j} - \xi_{x,\epsilon,\lambda}^{i,j}) \right)}
$$

$$
\geq \frac{\xi_{x,\epsilon}^{i,i} + \sum_{j \neq i} \left( \xi_{x,\epsilon,\lambda}^{i,j} + (\xi_{x,\epsilon}^{j,i} - \xi_{x,\epsilon,\lambda}^{j,i}) \right)}{\xi_{x,\epsilon}^{i,i} + 2\sum_{j \neq i} \left( \xi_{x,\epsilon,\lambda}^{i,j} + (\xi_{x,\epsilon}^{j,i} - \xi_{x,\epsilon,\lambda}^{j,i}) \right)} \geq \frac{1}{2}
$$

Since the above also holds for a sufficiently small neighborhood about $x$, we have the desired result.

$\square$

## E.2   PROOF OF THEOREM 3.5

Let us first recall the setting of Theorem 3.5, since it is more specialized than that of the previous results.

**Setting.** We consider the case of binary classification using a linear model $\theta^\top x$ on high-dimensional Gaussian data, which is a setting that arises naturally when training using Gaussian kernels. Specifically, we consider the dataset $\mathcal{X}$ to consist of $n$ points in $\mathbb{R}^d$ distributed according to $\mathcal{N}(0, I_d)$ with $d > n$ (to be made more precise shortly). We let the labels of points in $X$ be $\pm 1$ (so that the sign of $\theta^\top x$ is the classification), and use $X_1$ and $X_{-1}$ to denote the individual class points. Additionally, we let $n_1 = |X_1|$ and $n_2 = |X_{-1}|$.

Before introducing the proof of Theorem 3.5, we first present a lemma that will be necessary in the proof.

**Lemma E.1.** [Strict Convexity of $J_{mix}$ on Data Span] Suppose $n_1 = n_2 = 1$, i.e. there are two data points $x, z$ with opposite labels. If $x$ and $z$ are linearly independent, then $J_{mix}$ is strictly convex with respect to $\theta$ on the span of $x$ and $z$.

The reason this lemma focuses on the two data point case is that in the proof of Theorem 3.5 we will break up the Mixup loss into the sum over these cases. With this lemma, we may now prove Theorem 3.5. Before doing so, we point out that the version of $J_{mix}$ considered here is after composition with the logistic loss (since we are considering binary classification with a linear classifier).

**Theorem 3.5.** If the maximum margin solution for $X$ is also an interpolating solution for $X$, then any $\theta$ that lies in the span of $X$ and minimizes the Mixup loss $J_{mix}$ for a symmetric mixing distribution $\mathbb{P}_f$ is a rescaling of the maximum margin solution.

*Proof.* Since $d > n$, we have that all of the points $\{x_i\}_{i=1}^{n_1}$, $\{z_j\}_{j=1}^{n_2}$ are linearly independent with probability one. We will break the proof into two parts. In doing so, we make the following important observation: it suffices to prove the result for mixings of distinct points. This is because an interpolating solution (as given in Definition 3.3) is immediately seen to be optimal for the ERM part of $J_{mix}$ (the terms corresponding to mixing points with themselves). Thus, in what follows, we disclude these ERM terms from $J_{mix}$ to simplify the presentation.

**Part I:** $n_1 = n_2 = 1$. Denote $\theta^\top x_1 = u$ and $\theta^\top z_1 = -v$, then

$$J_{mix} = \mathbb{E}_\lambda \left[ \lambda \log(1 + \exp(-\lambda u + (1 - \lambda)v)) + (1 - \lambda) \log(1 + \exp(\lambda u - (1 - \lambda)v)) \right]. \quad (2)$$

Where $\lambda$ is distributed according to $\mathbb{P}_f$ which is symmetric and has full support on $[0, 1]$. Therefore, we can do the change of variables $\lambda := 1 - \lambda$, and

$$J_{mix} = \mathbb{E}_\lambda \left[ \lambda \log(1 + \exp(-\lambda v + (1 - \lambda)u)) + (1 - \lambda) \log(1 + \exp(\lambda v - (1 - \lambda)u)) \right]. \quad (3)$$

Combining Eq.(2) and Eq.(3), we know if $(u, -v)$ is a global minimum of $J_{mix}$, then so is $(v, -u)$. But the strict convexity in Lemma E.1 implies such a global minimum is unique, so we must have $u = v = k(\mathbb{P}_f)$, where $k(\mathbb{P}_f)$ is a constant that only depends on the density of $\mathbb{P}_f$. Furthermore, $\forall \lambda \in [0, 1]$, define

$$h_\lambda(k) = \lambda \log(1 + \exp((1 - 2\lambda)k)) + (1 - \lambda) \log(1 + \exp((2\lambda - 1)k)), \quad (4)$$

then $\forall k > 0$,

$$h_\lambda(-k) - h_\lambda(k) = (1 - 2\lambda) \log(1 + \exp((1 - 2\lambda)k)) + (2\lambda - 1) \log(1 + \exp((2\lambda - 1)k))$$

$$= (1 - 2\lambda) \log \left( \frac{1 + \exp((1 - 2\lambda)k)}{1 + \exp((2\lambda - 1)k)} \right) \geq 0, \quad (5)$$

and

$$h'_\lambda(k)|_{k=0} = -\frac{1}{2}(1 - 2\lambda)^2 \leq 0. \quad (6)$$

Hence, we must have $k(\mathbb{P}_f) > 0$.

**Part II: General $n_1$ and $n_2$.** For the general case, we extend the observation we made prior to the proof of Part I. Namely, if we can show that an interpolating solution is optimal for mixing across the two classes, it follows immediately that the solution is optimal for all of $J_{mix}$ (it is not hard to see that the calculation for mixing points from the same class is essentially no different from the ERM case in this context, as the $\lambda$ and $1 - \lambda$ terms can be combined). We thus focus only on mixing across classes, and overload the $J_{mix}^{i,j}$ notation to indicate mixing of points $x_i$ and $z_j$, so that we may write the loss in consideration as:

$$J_{mix}(\theta) = \frac{1}{n_1 n_2} \sum_{i=1}^{n_1} \sum_{j=1}^{n_2} J_{mix}^{i,j}(\theta). \tag{7}$$

By the proof in the previous part we know if $J_{mix}^{i,j}$ is minimized, then we must have $\theta^\top x_i = -\theta^\top z_j = k(\mathbb{P}_f) > 0$. On the other hand, if $J_{mix}^{i,j}(\theta)$ are minimized simultaneously for all pairs $(i, j)$, then clearly $J_{mix}(\theta)$ is also minimized. This is possible since the data points are linearly independent, so there exists $\theta \in \mathbb{R}^d$, such that

$$\theta^\top x_i = -\theta^\top z_j = k(\mathbb{P}_f) > 0 \quad \forall i \in [n_1], \ \forall j \in [n_2]. \tag{8}$$

Now we can conclude that any $\theta$ that minimizes the Mixup loss $J_{mix}$ is an interpolating solution. Restricting $\theta$ to the span of $\mathcal{X}$ finishes the proof. □

### E.2.1 PROOF OF SUPPORTING LEMMA

**Lemma E.1.** [Strict Convexity of $J_{mix}$ on Data Span] Suppose $n_1 = n_2 = 1$, i.e. there are two data points $x, z$ with opposite labels. If $x$ and $z$ are linearly independent, then $J_{mix}$ is strictly convex with respect to $\theta$ on the span of $x$ and $z$.

*Proof.* We note again that it suffices to prove the strict convexity with respect to only the mixings of different points, as the ERM part is clearly strictly convex and the sum of two strictly convex functions remains strictly convex. Denote

$$f(\lambda) = \lambda x + (1 - \lambda)z, \tag{9}$$

then $J_{mix}$ can be expressed as

$$J_{mix}(\theta) = \mathbb{E}_\lambda \left[ \lambda \log\left(1 + \exp\left(-\theta^\top f(\lambda)\right)\right) + (1 - \lambda) \log\left(1 + \exp\{\theta^\top f(\lambda)\}\right) \right]$$
$$= \mathbb{E}_\lambda \left[ \log\left(1 + \exp\left(-\theta^\top f(\lambda)\right)\right) + (1 - \lambda)\theta^\top f(\lambda) \right]. \tag{10}$$

Where again $\lambda \sim \mathbb{P}_f$. Note that the second term in Eq.(10) is linear in $\theta$, hence the Hessian of $J_{mix}$ can be written as

$$\nabla^2 J_{mix}(\theta) = \mathbb{E}_\lambda \left[ \frac{\exp\left(-\theta^\top f(\lambda)\right)}{(1 + \exp(-\theta^\top f(\lambda)))^2} f(\lambda)f(\lambda)^\top \right] \tag{11}$$
$$:= \mathbb{E}_\lambda g(\lambda).$$

Define $\mathcal{B} := \text{Span}\{x, z\}$. To show $J_{mix}$ is strictly convex on $\mathcal{B}$, it suffices to show for every non-zero vector $a \in \mathcal{B}$, we always have

$$a^\top \nabla^2 J_{mix}(\theta) a > 0. \tag{12}$$

Note that $g(\lambda)$ is continuous *w.r.t.* $\lambda$ and that $\mathbb{P}_f$ has full support on $[0, 1]$, it suffices to show either

$$a^\top f(0)f(0)^\top a > 0 \ \text{ or } \ a^\top f(1)f(1)^\top a > 0, \tag{13}$$

which is equivalent to either

$$a^\top x \neq 0 \ \text{ or } \ a^\top z \neq 0. \tag{14}$$

This is obvious since $a$ is a non-zero vector in $\mathcal{B}$, and that $x$ and $z$ are linearly independent. □

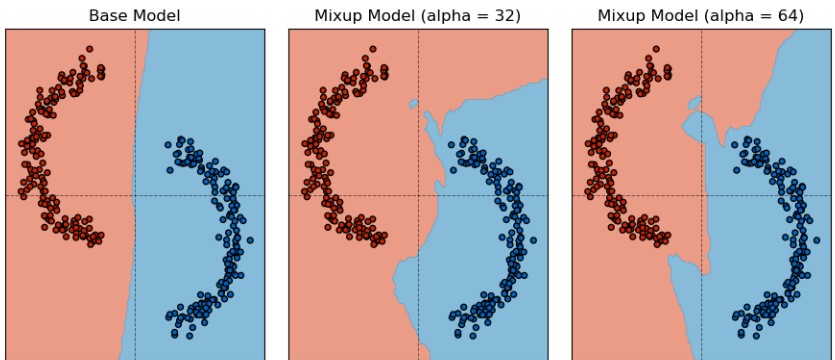

Figure 10: Decision boundary plots for $\alpha = 32,\ 64$ and a class separation of 0.5.

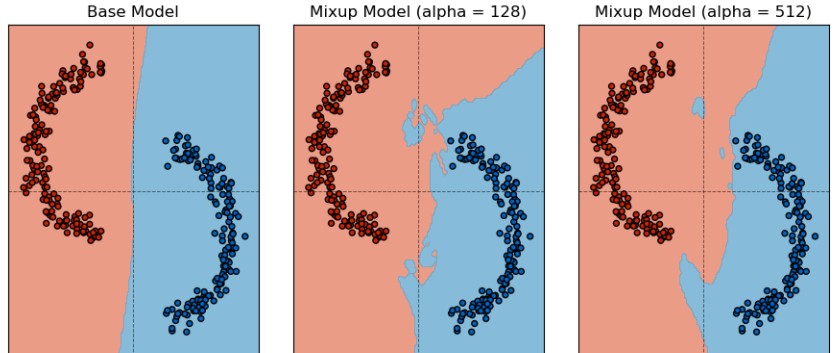

Figure 11: Decision boundary plots for $\alpha = 128,\ 512$ and a class separation of 0.5.

## F    ADDITIONAL EXPERIMENTS FOR SECTION 3

In this section, we consider different class separations and choices of the mixing parameter $\alpha$ when training on the two moons dataset, with all other experimental settings being the same as in Section 3.1.

Upon decreasing the class separation to 0.1, we note that even standard training captures more of the nonlinear aspects of the data, as was observed in the prior work of Pezeshki et al. (2020).

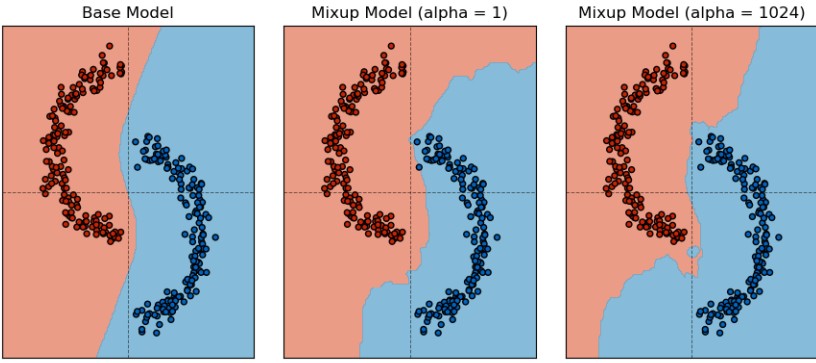

Figure 12: Decision boundary plots for $\alpha = 1,\ 1024$ and a class separation of 0.1.

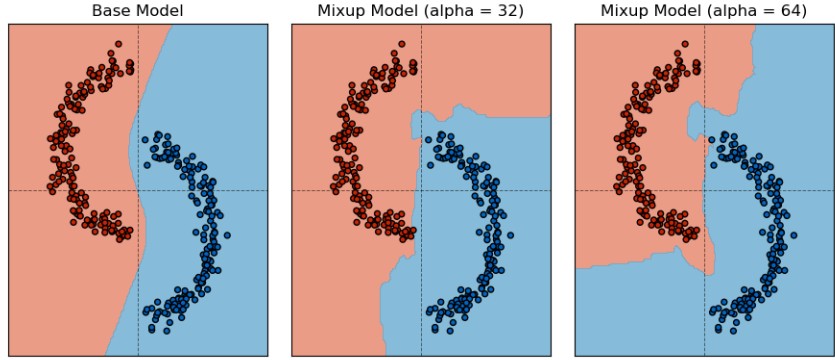

Figure 13: Decision boundary plots for $\alpha = 32,\ 64$ and a class separation of 0.1.

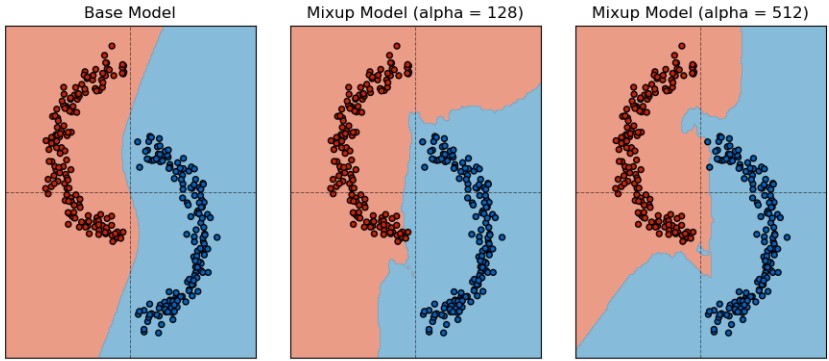

Figure 14: Decision boundary plots for $\alpha = 128,\ 512$ and a class separation of 0.1.

