# OpenReview forum: "Towards Understanding the Data Dependency of Mixup-style Training"
_ICLR.cc/2022/Conference — ICLR 2022 Spotlight_

### Official Review · Reviewer_NoVN · 2021-11-01

**Correctness:** 3
**Technical Novelty And Significance:** 3
**Empirical Novelty And Significance:** 3
**Recommendation:** 6
**Confidence:** 4

**Main Review:**

The paper provides an interesting  study, from the properties of the training data perspective, why and when Mixup help training better classifier in terms of model generalization and adversarial robustness. The papers show that both the model generalization and adversarial robustness brought by Mixup training heavily depends on the properties of the training data. Such observations have been empirically observed (such as by Guo et al. in (2019) as cited), but this paper provides a more formal theoretical analysis on that. I think this paper contribute an interesting analytical framework and useful insights regarding Mixup training to the community. Nevertheless, I have the following comments to the paper.

1.	The paper did construct a case that Mixup failed to minimize the original empirical risk, and conjectured (at the end of section 2.3) that datasets with collinear structures amongst data points may cause Mixup to fail. I wonder if the authors could be more precise and specific on the conditions for such Mixup failure and formally prove that to some extent.
2.	For figure 2, I wonder if the Mixup training errors are on the original training points or on the mixed points. It seems to me that Mixup obtained higher training error than the training without Mixup. Using these observations to support that “mixup training minimizes the empirical risk” (the first sentence of the last paragraph in Page6) is not very convincing to me. What training errors would you get if you train the models with a smaller \alpha?
3.	I do not fully understand the argument in the last two sentences of Section2.4: “…This is in contrast to the fact that the test data points exhibit greater angular distance to the mixed training…”. To me, the reason for the test performance of Mixup at \alpha =1024 is significantly worsening than ERM could be that the testing points are very different from the mixed points even though the training loss is low. Also, the manifold intrusion issue as discussed in Guo et al. (2019) seems to indicate the cases that mixed samples collide with the original training samples, not the test points. In your case, Mixup training with \alpha = 1024 seems to not having such intrusion issue, but augmented samples generated by Mixup may have the affinity issue (i.e., far away from the training set). See “Tradeoffs in Data Augmentation: An Empirical Study” (Gontijo-Lopes et al., ICLR 2020).
4.	Section 2.5 is interesting to me, but also remind me of a recent paper “Midpoint Regularization: From High Uncertainty Training Labels to Conservative Classification Decisions” (Guo, ECML 2021). In that paper, mid-point samples (a special setting of Mixup) are generated for training. The empirical studies in that paper seems to be consistent to your theorem 2.11 here. In this sense, further discussion relating to the observations in that paper could be beneficial.
5.	The paper could benefit from a more detailed related work section for the completeness of the paper. For example, Mixup has been applied to other settings such as natural text and graph data. Another missing work is “Mixup as Directional Adversarial Training” (Archambault et. al., 2019), which also attempts to provide a theoretically framework to understand why Mixup works, and their analysis seems to be related to section 3 in this paper.


Minor question:

The theoretical analysis here is under the setting that the mixing ratio \lambda is the same for mixing both the inputs and labels. Recent work on Mixup which uses different mixing ratios for the input and label mixing seems also working well (e.g., "Nonlinear Mixup: Out-Of-Manifold Data Augmentation for Text Classification", Guo, AAAI2020). I am curious to know if your theory in the paper still holds in that setting.


**Summary Of The Paper:**

The paper provides a theoretical framework to enable analyzing Mixup training from a data dependency perspective, aiming to understand how the structure of the training data impacts the effectiveness of Mixup training. The paper first shows that the benefits of Mixup training, in terms of model generalization and adversarial robustness, heavily rely on the properties of the training data, and studies when Mixup training can fail to minimize the original empirical risk. In addition, the paper also theoretical characterizes the margin of a Mixup classifier, justifying why a Mixup classifier’s decision boundary is better than standard training and when such advantage disappear (i.e., in cases of training Mixup with linear models and linearly separable datasets). The paper contributes an interesting perspective for looking into Mixup training, which has been widely adopted by the community.

**Summary Of The Review:**

The paper contributes to the understanding of Mixup training from a training data dependency perspective, which is novel and I think it would benefit the research community. On the other hand, the paper is not clear to me in some places as detailed in my reviews, and I would like the authors to comment on them.

---

> ### Author Response · Authors · 2021-11-11
> **Response to Reviewer NoVN**
>
> We would like to thank Reviewer 3 (NoVN) for taking the time to review our paper. We appreciate that they found our work interesting, and we are grateful for the additional useful references that they pointed out (which have been included). We hope to address all points in the review below, in the order in which they were made.
>
> 1. It is difficult to be more precise for Mixup failure without making explicit assumptions about the dataset and mixing distribution (like we have done in Section 2.3). Basically, the idea is that when you have collinear structure like in our example dataset **and** a mixing distribution that ends up concentrating where the collinearity happens (in our case, midpoints), then Mixup can fail as per our analysis. The general practical advice we give is bundled with our sufficiency conditions proved in Section 2.4, where we point out that when this type of collinear structure is not present then Mixup will work on the original data *regardless* of the mixing distribution.
>
> 2. In Figure 2, the training error is on the original data points for both Mixup and ERM - thank you for bringing this up, as we have clarified it in the revision. It is true that Mixup achieves higher training error than ERM, but only very marginally so (within 1\%). Furthermore, the reason we used $\alpha = 1024$ in our experiments was to illustrate that even when the mixing distribution moves very far from the ERM regime, Mixup still essentially minimizes the original risk when collinearity is not present (we are basically training on midpoints here - and only for 50 epochs - but Mixup is getting only slightly higher training error than ERM). In a sense, $\alpha = 1024$ is a "worst case". We have included additional experiments for $\alpha = 1, 32, 128$ in the Appendix to show that for these smaller values of $\alpha$, the training error curves are even closer between Mixup and ERM. Hopefully this alleviates any concerns that $\alpha = 1024$ was cherry-picked.
>
> 3. We agree that the discussion at the end of Section 2.4 was quite confusing as originally written, and we have reorganized it significantly to hopefully make it clearer. Regarding the comparison to Guo et al. (2019), we believe Figure 1 in their paper is actually of test error (although it is not labeled at all), because the numbers are simply too high to be training error for ResNet on CIFAR. The point we were trying to make was that in their paper they claim that their manifold intrusion phenomenon occurs for MNIST and CIFAR-100, but our experimental results and approximate collinearity actually contradict this. As can be seen from our plots for $\alpha = 1, 32, 128, 1024$, Mixup achieves near perfect training accuracy on MNIST, CIFAR-10, and CIFAR-100 on the original unmixed data. We totally agree that one explanation for degraded test performance with Mixup is due to the potential lack of "affinity" between mixed points and test points as you point out, we just wanted to say that this degradation cannot necessarily be explained by manifold intrusion/collinearity ideas (since we find that they are not present).
>
> 4. Thank you for this reference, and we apologize for having missed it in the original version of our paper. We now point out in Section 2.5 that the phenomenon that we are discussing has also been observed in practice by Guo et al. (2021).
>
> 5. Agreed, we have expanded the related work with some suggestions from Reviewer 2 (eGEK) and have added some of the references you mention as well. We now also mention the related work of Archambault et. al. (2019) in Section 3.2, thank you for pointing it out to us.
>
> Regarding the minor question - the idea of using a different mixing distribution for labels is definitely interesting. We believe our theoretical framework can definitely be adapted to this setting by introducing another mixing distribution (so in addition to $P_f$ we will have a $P_{\ell}$ for labels), and that analogues of the results that we prove should hold. For example, the collinearity issue can arise regardless of the label mixing distribution (i.e. our example in Section 2.3 will still work).

---

> > ### Comment · Reviewer_NoVN · 2021-11-29
> > **Thank you and no further questions**
> >
> > I appreciate the authors for answering  my raised questions in detail.
> > The revision at the end of Section 2.4 is now concise and clear to me.
> >
> > I have no other questions at this point and will discuss the paper with other reviewers and ACs.

---

### Official Review · Reviewer_eGEK · 2021-11-02

**Correctness:** 3
**Technical Novelty And Significance:** 3
**Empirical Novelty And Significance:** 3
**Recommendation:** 8
**Confidence:** 2

**Main Review:**

# Strengths:

* The approach the paper takes in their study is a meaningful one, and the theory and experiments are well motivated.

* The paper goes beyond the observations of related works to provide concrete conditions on which Mixup training can fail, and demonstrate it by constructing synthetic datasets and measures that could be estimated for natural datasets. This could be useful to practitioners as well for when they design their experiments on natural data.

* The paper takes a complementary approach to existing works for understanding the generalization and robustness of the Mixup-optimal classifer.

* Some interesting future work directions are underscored by the paper, such as the rate of convergence of Mixup training (Section 2.5).

# Weaknesses:

* `Mixup at our choice of $\alpha=1024$`: could you report your experiments for better context? Furthermore, could you explain what the argument is in this paragraph? How does the observation that the test data points exhibit greater angular distance connect with your empirical observation? Do you mean to say that greater angular distance for the test set means less "collision" that is discussed in Guo et al. (2019)? I think clarification of this paragraph will improve the paper, because it is at the transition between studying the Mixup optimal classifier and its generalization properties.

* I am curious what kind of applications of your Sufficient Conditions you envision for experiments. Could you elaborate on some interesting directions or limitations as you try to scale to larger and higher-dimensional raw data?

* The Proof Sketch of Lemma 2.3. is unclear to me. Is it correct to say that you "define" $h_\epsilon$ to be the argmin? Do you mean that you "construct $h_\epsilon$ and show it is the argmin" instead?

* The Proof Sketch of Proposition 2.8 is unclear to me. How do we know that the constant measure gives the empirical risk minimization?

* Isn't $\epsilon$ missing in your definition of $\xi_{x,\epsilon,\lambda}^{i,j}$?

* Not clear what you mean by "recover" in Theorem 2.11. Do you mean recover as a linear combination? Could you clarify it in the paper?

* In Section 3.1. Paragraph "Experiments" exactly to what does *this* refer to in "To illustrate this"? Could you clarify the connection with the previous paragraph?

* Not sure about the summation indices in the proof of Lemma 2.3. In the definition of $J_{mix}$ I think the summation goes only from $j=i+1$ to $k$. How does the summation (and the sign of the summation) change to indices $j=1$ to $i-1$ (and a negative sign) in the last summand in the equation at the end of page 15?


* Do you know of other papers that try to find examples for when the Mixup optimal classifier coincides with the empirical cross-entropy minimizer?


## Minor.

* Suggestion for the Introduction. Mixup has been used in contrastive learning settings. You could consider citing some of the works, e.g. [1,2].

* Definition of J_mix on top of page 3. I would suggest to make the dependence on $g$ explicit in the RHS.

* On page 3 you could mention that $A_{x,\epsilon,\delta}^{i,j}$ would be used later in the paper.

* Lemma 2.3. Is it that $\epsilon \to 0$? You could clarify this.

* Figure 1 and 8: ERM curves do not depend on $\alpha$? Or am I missing something? Could you clarify this in the captions. What causes the difference between the blue (ERM) curves in Fig. 8(a) and (b)?

* Figure 1's labeling is not clear. Please, enlarge the font.

* Definition 8 is actually Figure 8 in Section D. Maybe a typo in the link?

* The abuse of notation in Proposition 2.2. make it hard to understand the proof.

* In the proof of Theorem 2.11, I am not sure what you mean by $Aw^*=PAw$ is equivalent to $[A, PA][(w^*)^\top, w^\top]^\top$. Could you clarify?

# References

[1] Lee et al. 2021. i-Mix: A Domain-Agnostic Strategy for Contrastive Representation Learning. In ICLR.

[2] Verma et al. 2021. Towards Domain-Agnostic Contrastive Learning. In ICML.


**Summary Of The Paper:**

This paper provides interesting theoretical tools that help us study the relationship between empirical risk minimization/ empirical cross-entropy minimzation and Mixup minimization. The contributions of the paper consist of the following:

1. Constructing a dataset on which Mixup training fails to obtain the empirical risk minimization.
2. Providing sufficient conditions for minimizing the original risk and estimates of these sufficient condition measures on natural datasets.
3. An observation that the rate of empirical risk minimization using Mixup and a theorem about why this should happen.
4. A study on when the Mixup optimial classifier fails to generalize and a demonstration on a toy dataset.
5. An example for when Mixup training leads to the same classifier, obtained from standard training.

The main difference with related works is that the paper attempts to understand *why* Mixup works by studying *when* Mixup works with concrete examples.



**Summary Of The Review:**

I recommend 8: accept, good paper. In my opinion, this paper will be useful to the community both for theory and experiments. I have listed a few questions and suggestions in the Main Review. It is possible that I have missed some of the main arguments in the related works, but to me the contributions of the paper are novel and useful.

---

> ### Author Response · Authors · 2021-11-11
> **Response to Reviewer eGEK (Part 1)**
>
> We would like to thank Reviewer 2 (eGEK) for taking the time to review our paper so thoroughly. We truly appreciate the care with which this review was written, and we are grateful to the reviewer for combing through not only the main body of the paper but also the supplementary material. We hope to address all points in the review below (in the order in which they were made).
>
> **Strengths:** Thank you for the kind words.
>
> **Weaknesses:**
> - We totally agree that the discussion of how to apply our sufficiency conditions is somewhat confusing upon rereading. We have reorganized that part of Section 2.4 to hopefully make things much clearer. The practical take-away is that, when datasets do not exhibit approximate collinearity between points of different classes (where the "approximate" depends on the Lipschitz constant of the model being considered), Mixup should perform virtually the same as ERM with regards to error on the original training data. We hope that in the rewritten version, both our experimental setup and verification of approximate collinearity are more easily understood. Additionally, we have supplemented our $\alpha = 1024$ experiments with experiments for $\alpha = 1, 32, 128$ to address some concerns that $\alpha = 1024$ was cherry-picked to illustrate our point (in fact it's the opposite, we put $\alpha = 1024$ in the main body to show that even when you move very far from the ERM regime Mixup still minimizes the original risk when collinearity is not present, as our theory predicts). Finally, regarding your question concerning Guo et al. (2019), the answer is yes - basically in their paper they suggest that true point/Mixup point collisions are the reason for performance degradation with Mixup, but our experiments indicate otherwise as the distance values are quite large (for both train and test data). Note also that in reorganizing this subsection we swapped the angular distance values for Euclidean distance values, both for the easier interpretability with respect to the Lipschitz constant of the networks (angular distance being large implies Euclidean distance is large) and because our previous line concerning positive homogeneity of ReLUs is not entirely accurate justification for angular distance (since there are still biases in addition to the weights in the network).
> - The main use case we envision for the sufficiency conditions is essentially to check for approximate collinearity in data with a procedure similar to the one we take in the paper, as the absence of such collinearity should indicate that Mixup will work. We now highlight this "practical takeaway" in the experimental discussion in Section 2.4.
> - Essentially, yes - we take $h_{\epsilon}$ to be the argmin but it is not clear a priori that this is even a well-defined function of $x$ (since the argmin need not be unique in general). In the proof we show that it is, and in fact this function is continuous with its coordinate functions exhibiting the form shown in Lemma 2.3. We have updated the proof sketch to hopefully make all of this more clear.
> - In Proposition 2.8, the proof sketch is meant to indicate that when you consider $\lim_{\epsilon \to 0} h_{\epsilon}^i(x)$ (with $x$ in the support of class $i$), all of the non-same-point mixing terms in $h_{\epsilon}^i(x)$ vanish. This is because we assumed the mixing distribution to be continuous, so mixing two points that are not exactly $x$ continuously cannot assign positive measure to the single point $x$. On the other hand, mixing $x$ with itself of course can only give you $x$. We realize upon rereading that this may definitely be a leap in logic given what is presented in the sketch, so we've added a bit more there to hopefully alleviate this issue.
> - Thank you for catching this; $\epsilon$ was indeed missing in the definition of $\xi_{x, \epsilon, \lambda}^{i, j}$, this has been fixed (it just needed to be added in the subscript of the integration region).
> - Sorry for being unclear in Theorem 2.11 - we meant recover as in uniquely determine the original points from their midpoints. The wording has been updated here.
> - Our apologies for the unclear antecedent in the "Experiments" part of Section 3.1 - we have clarified to indicate that we were referring to the fact that Mixup can lead to a nonlinear decision boundary even when regular training does not.
> - Sorry, upon rereading we see now that it is not a trivial jump to make from "unpacking $J_{mix}$" to the form we've written in the proof. Basically, the summation from $j = 1$ to $j = i-1$ is from collecting terms (we are organizing each of the $\log \theta^i$ together in the summation). The negative sign in front of the combined summation (second term) is a typo though, thank you very much for catching that. It should now be more straightforward to check that doing the Lagrange multiplier calculation (optimizing the updated expression with the constraint that $\sum \theta^i = 1$) yields the form written in Lemma 2.1.

---

> > ### Author Response · Authors · 2021-11-11
> > **Response to Reviewer eGEK (Part 2)**
> >
> > **Weaknesses (Continued):**
> > - In our original survey of the related work we did not find any work that contained results similar to the ones we have in Section 3.2. However, as reviewer NoVN pointed out to us, the work of Archambault et al. (2019) does contain some related ideas (when Mixup coincides with a certain, very similar form of adversarial training) which we have now referenced at the beginning of Section 3.2.
> >
> > **Minor Issues:**
> > - Thank you for the additional references; we have included them in the introduction.
> > - We have added the dependence on $g$ to the RHS of the definition of $J_{mix}$, we agree this is clearer notation.
> > - We have added that $A_{x, \epsilon, \delta}^{i, j}$ will be used in Section 3 after introducing it.
> > - Right, in Lemma 2.3 it's $\epsilon \to 0$ and you are correct - that should be clarified. It is now fixed, thank you for pointing this out.
> > - The difference between the blue curves is largely due to the different scalings of the $y$-axis for different choices of $\alpha$, and partially also because the random seeds were not fixed to be the same between those runs (the seed were not fixed a priori in our experiments to verify that the observed behavior is independent of some lucky/unlucky initializations).
> > - The labeling in both Figures 1 and 2 has been enlarged, sorry for this oversight.
> > - Definition 8 is a typo, it should be referencing Definition 2.1 and this has been fixed now (that was indeed a link problem).
> > - Agreed, the abuse of notation in the proof of Proposition 2.2 is perhaps a bit much without additional exposition; we have added some more explanation which hopefully clarifies the notation and the proof.
> > - In the proof of Theorem 2.11, we accidentally omitted the definition of the $[A, B]$ notation, which was intended to mean concatenation of columns. This has been clarified, and the line you pointed out has been fixed with some additional explanation.
> >
> > Finally, we'd once again like to thank you for being so diligent in your review and catching all of the slight inaccuracies mentioned above that are now fixed.

---

### Official Review · Reviewer_mn55 · 2021-11-03

**Correctness:** 3
**Technical Novelty And Significance:** 2
**Empirical Novelty And Significance:** 2
**Recommendation:** 3
**Confidence:** 3

**Main Review:**

I like the topic of this paper, but I have several concerns as below.

1. Focus on extremely high \alpha values
    1. In the illustration of mixup examples, this paper considers \alpha=32, 128, 1024, which is not even considered in the original mixup paper (standard one is \alpha=1, and maybe variant the authors considered is \alpha=2). My personal feeling is, no one will use such large alpha in real application of mixup; they will just use default value \alpha=1. I think the authors are setting extremely large \alpha values to exaggerate the effect they want to show.
2. Theory results are limited to simple settings
    1. Proposition 2.5 and Theorem 2.7 holds for only a certain dataset. I guess both are not deserved for proposition & theorem, just some “example”.
    2. Results on Sec.3.2 are applicable for linear model with Gaussian data
3. We cannot know whether the assumptions are realistic
    1. Assumption 2.9 and 3.1 -> How can we know that this assumption holds in real datasets?
4. Conclusion of Sec.2.5 is unclear
    1. It seems like the Sec.2.5 should be about analyzing the convergence rate of original empirical risk when we use mixup training. Is Theorem 2.11 sufficient for showing that theoretically? I guess not. Moreover, the “midpoints” discussed here is mixing with \lambda=0.5, which maps to an extremely large \alpha value case, which is unrealistic.


**Summary Of The Paper:**

This paper analyzes the empirical risk of mixup-style training methods. First, the authors provide an example where mixup training cannot minimize the empirical risk on the original data. Second, the authors provide sufficient conditions for mixup to minimize the empirical risk.


**Summary Of The Review:**

This is an interesting paper trying to answer “why” and “when” mixup works. But it is neither a theory paper providing some meaningful theoretical results (that can be applicable for general scenarios), nor an empirical paper suggesting new scheme and providing extensive experimental results. I would say this is a toy theoretical attempt, which is not suitable for ICLR acceptance.

---

> ### Author Response · Authors · 2021-11-09
> **Response to Reviewer mn55**
>
> We would like to thank Reviewer mn55 for taking the time to review our paper. We believe that all of the claims made in this review have been addressed in the originally submitted version of the paper, and we hope to clarify this below.
>
> 1. Extremely high $\alpha$ values
>     1. In all of the experiments in our work, we consider a wide band of $\alpha$ values when mixing using $\mathrm{Beta}(\alpha, \alpha)$ (ranging from, as pointed out, 1 to 1024). It is not true that in practice people only use small values of $\alpha$; in fact, this is not even true in the original paper as suggested in the review. Please see Section 3.5 in the original paper [1], where $\alpha = 32$ was the most effective choice for robustness to label noise when not combining Mixup with dropout.
> Additionally, it is inaccurate to say that we are focusing on extremely large values of $\alpha$ to exaggerate our results. In Section 2.3 of our work (where we construct a Mixup failure case), we show a failure of Mixup with $\alpha = 32$ and then point out that we can make this failure happen even for $\alpha = 1$ (for which we include experiments in the supplementary material) by simply adding more data points to our construction. We opted not to present this in the main body to simplify the theoretical presentation, which is most straightforward in the minimalistic 3 data point setting we present.
> Furthermore, the choice of $\alpha = 1024$ in the experiments in Section 2.4 was made to indicate that even when mixing with a seemingly unrealistic mixing distribution (far from the ERM choice of $\alpha = 0$), we still minimize the empirical risk (the original loss) when our sufficient conditions approximately hold. The motivation for this is actually the opposite of what is suggested in the review - it is not to exaggerate our results in any way, but rather to show that they can explain some of the phenomena that occur even for unreasonable mixing distributions like the ones suggested in our counterexample construction (instead of say, presenting the same plots for very mild values of $\alpha$ where we already know from prior works that the original risk is also minimized).
> 2. Theory limited to simple settings
>     1. Proposition 2.5 and Theorem 2.7 do indeed only hold for specific datasets as written, but that is because they are meant to illustrate general ideas. One can of course construct a dataset of arbitrary complexity by having $N - 3$ arbitrary points in addition to the 3-point dataset we discuss in Theorem 2.7; this will still lead to Mixup failing to minimize the original risk, but its failure will be proportional to $N$. Almost all of the other theoretical results in our paper actually hold, as we point out, for arbitrary continuous mixing distributions supported on $[0, 1]$, so we do not believe we are limiting ourselves to simple settings in any way.
>     2. While it is possible to consider linear models with Gaussian featurization a "simple setting", we respectfully disagree - this has been a common assumption in many theoretical work that nevertheless gave useful insights. For more background, we recommend the reviewer read the discussion in [2], from which we get our Lemma 3.7. Our theorem can also be extended to other settings where all points are support vectors, which is more general than Gaussian featurization (see Section 1.1 in [2], where they show empirically that this occurs also for Fourier features).
> 3. We cannot know whether the assumptions (2.9 and 3.1) are realistic
>    1. The entirety of the second half of Section 2.4 is dedicated to verifying that Assumption 2.9 holds on realistic datasets (MNIST, CIFAR10, CIFAR100). We include a detailed description of how we approximately check this assumption, and also how Mixup training on these datasets minimizes the empirical risk even with an unreasonable mixing distribution (Figure 2). We definitely acknowledge that some of the writing in this section can be made clearer, as pointed out by the other reviewers, and we will make these clarifications when we upload a revision of the paper shortly. As for Assumption 3.1, we in fact point out immediately that this assumption holds for all points in the support of each class, which is how we derive Theorem 2.10 from Theorem 3.2.
> 4. Conclusion of Section 2.5 is unclear
>    1. We point in Section 2.5 that Theorem 2.11 is purely an information-theoretic result; we are simply showing that it is possible to recover the original points and their labels from only the mixed points and the mixed labels. As pointed out in the review, midpoint mixing is "unrealistic", which is exactly why this result is interesting - it is in fact still possible to get the correct classifications on the original data from this unrealistic mixing. As far as a convergence analysis is concerned, we also mention that such an analysis is outside of the scope of this work, but would be interesting for future work.
>
> [1] arXiv:1710.09412v2
>
> [2] arXiv:2005.08054v2

---

### Author Response · Authors · 2021-11-11
**Updates in Revision**

We would like to thank the reviewers for their many helpful comments. We have revised our paper to hopefully address all of them, and we point out the main changes below for convenience.

1. The related work has been expanded to incorporate all of the suggested missed references. Furthermore, we have included some additional references in Sections 2.5 and 3.2 as per the suggestion of reviewer NoVN.

2. The part of Section 2.4 discussing how to apply our sufficiency conditions has been reorganized to hopefully be much clearer.

3. Additional experiments using different mixing parameter choices have been added to complement the experiments in Section 2.4. Mainly, these have been added to show that the only thing particular about $\alpha = 1024$ in Section 2.4 is that it is in some sense a "worst case" analysis - Mixup still gets near-perfect training accuracy on the original data when training on midpoints. The results for smaller values of $\alpha$ are strictly better in terms of how similar Mixup is to ERM in terms of original training error.

4. We have enlarged the text in the graphs in Figures 1 and 2, we apologize for the inconvenience caused by the originally small text. In addition, we have regenerated all of the plots to fix a minor inaccuracy in the displayed error bars (and in the process updated Table 1, as the random seeds were not fixed before - the change is very marginal).

5. All of the typos, minor omissions, and unclear explanations pointed out by reviewer eGEK have been fixed. We hope now that the proofs in the supplementary material are clearer. We would also like to reiterate how grateful we are to reviewer eGEK for going through our paper so carefully.

---

### Decision · Program_Chairs · 2022-01-20

**Decision:**

Accept (Spotlight)

**Comment:**

This paper presents an interesting analysis of mixup, discussing when it works and when it fails. The theory is further illustrated with small but intuitive examples, which facilitates understanding the underlying phenomena and verifies correctness of the predictions made by the theory. The submission has received three reviews with high variance ranging from 3 to 8: mn55 favoring rejection while eGEK recommending accept. I read all the reviews and authors' response. Unfortunately, mn55 did not follow up to express how convinced they are with author's reply, but I do find the responses to mn55 very solid and convincing. In concordance with eGEK, I do find the provided analysis important and helpful, and the presentation of the theory through concrete examples very compelling.